# OptimalThinkingBench: Evaluating Over and Underthinking in LLMs

**Pranjal Aggarwal**[*]
FAIR at Meta & Carnegie Mellon University

**Seungone Kim**
FAIR at Meta & Carnegie Mellon University

**Jack Lanchantin**
FAIR at Meta

**Sean Welleck**
Carnegie Mellon University

**Jason Weston**
FAIR at Meta

**Ilia Kulikov**
FAIR at Meta

**Swarnadeep Saha**[†]
FAIR at Meta

## Abstract

Thinking LLMs solve complex tasks at the expense of increased compute and overthinking on simpler problems, while non-thinking LLMs are faster and cheaper but underthink on harder reasoning problems. This has led to the development of separate thinking and non-thinking LLM variants, leaving the onus of selecting the optimal model for each query on the end user. We introduce `OptimalThinkingBench`, a unified benchmark that jointly evaluates overthinking and underthinking in LLMs and also encourages the development of optimally-thinking models that balance performance and efficiency. Our benchmark comprises two sub-benchmarks: `OverthinkingBench`, featuring simple math and general queries in 72 domains, and `UnderthinkingBench`, containing 11 challenging reasoning tasks along with harder math problems. Using novel thinking-adjusted accuracy metrics, we extensively evaluate 33 different thinking and non-thinking models and show that no model is able to optimally think on our benchmark. Thinking models often overthink for hundreds of tokens on the simplest user queries without improving performance. In contrast, large non-thinking models underthink, often falling short of much smaller thinking models. We further explore several methods to encourage optimal thinking, but find that these approaches often improve on one sub-benchmark at the expense of the other, highlighting the need for better unified and optimal models in the future.[1]

## 1 Introduction

Users query LLMs across a spectrum of tasks from factual queries to code and math proofs, so a useful LLM should answer easy questions quickly while spending more time on harder ones for better accuracy. In the past, LLMs have performed well on easy problems but have underthought on complex reasoning problems that required step-by-step thinking (Wei et al., 2022). In contrast, recent "thinking" LLMs (DeepSeek-AI et al., 2025; OpenAI et al., 2024) markedly improved the latter (Muennighoff et al., 2025; Aggarwal & Welleck, 2025) but at the cost of overthinking on simple tasks, harming latency, cost, and in some cases even performance (Cuadron et al., 2025; Chen et al., 2025a; Gema et al., 2025; Liu et al., 2025). Consequently, many state-of-the-art LLMs have separate thinking and non-thinking variants, forcing end-users to manually decide which model is best – an unrealistic requirement for optimal accuracy-efficiency trade-off at scale. To encourage the development of such *optimally-thinking models* that balance cost and performance,

---

[*]Equal contribution. Correspondence: `pranjal2041@gmail.com`

[†]Correspondence: `swarnadeep@meta.com`

[1]Code is released at: `https://github.com/facebookresearch/RAM/tree/main/projects/otb`. Data released at: `https://huggingface.co/datasets/facebook/optimal_thinking_bench`

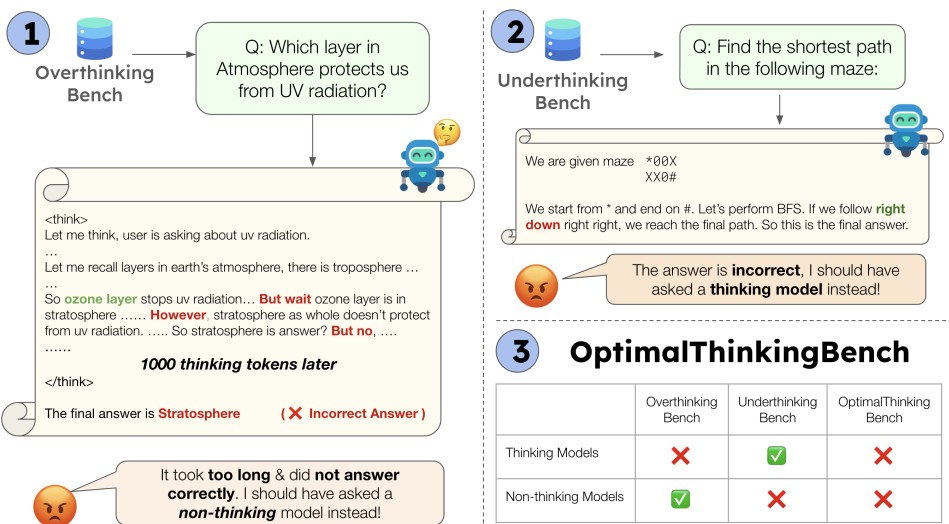

Figure 1: `OptimalThinkingBench`: A unified benchmark to evaluate overthinking and underthinking in LLMs. `OverthinkingBench` consists of simpler queries where excessive thinking either does not improve or occasionally degrades performance. `UnderthinkingBench` consists of reasoning problems where lack of thinking hurts performance.

we introduce a new benchmark called `OptimalThinkingBench`. It is a combination of two new sub-benchmarks: `OverthinkingBench` and `UnderthinkingBench` that allows us to evaluate and develop methods for *optimal* reasoning across a wide variety of domains.

To first address the challenge of overthinking by thinking models, we introduce `OverthinkingBench`, a benchmark containing simple queries where non-thinking models achieve high accuracy but thinking models yield similar or even lower scores despite generating hundreds of thinking tokens. We synthetically construct `OverthinkingBench` with automated filtering that ensures difficulty control, disambiguation, and answer correctness. It consists of both general and mathematical questions across more than 72 domains with four distinct answer types. We then introduce `UnderthinkingBench`, consisting of 11 challenging reasoning tasks from 6 different domains (games, algorithms, graphs, arithmetic, geometry, and logic) (Stojanovski et al., 2025), along with 2 competition math benchmarks. It is constructed based on the principle that for certain questions, no matter how large a non-thinking model is, its performance on complex reasoning tasks will be lower than that of a much *smaller yet thinking* model. Taken together, the synthetic components of both sub-benchmarks allow each to remain dynamic, ensuring that the data generation recipe can be used to prevent benchmark contamination and evolve with increasing model competence. See Figure 1 for two example queries from our benchmark.

To track progress on `OverthinkingBench`, we first propose Overthinking-Adjusted Accuracy (OAA), a metric that computes sample correctness below a certain thinking budget. We use this metric to calculate $AUC_{OAA}$, an overthinking measure computing the area under the OAA curve to account for a range of such thinking budgets. Our final metric for `OptimalThinkingBench` is the $F_1$ score between the overthinking $AUC_{OAA}$ and the underthinking accuracy.

We perform comprehensive evaluations with 33 different models to show that current thinking models overthink even on simple queries without improving performance, leading to a substantial drop in user experience and increasing cost. Non-thinking models, on the other hand, underthink on difficult reasoning problems. Notably, *no single model can optimally balance accuracy and efficiency* on our benchmark, highlighting the importance of our benchmark. Finally, we explore various training-time as well as test-time approaches to optimal thinking that rely on reward shaping, routers, or deliberate prompting. We also analyze both qualitatively and quantitatively how models overthink and underthink across different domains and answer types. Our results indicate that while some of these methods prove to be more effective than others, a significant gap persists, which motivates the need for better optimally-thinking LLMs in the future.

In summary, our contributions are three-fold: First, we develop `OptimalThinkingBench`, a single unified benchmark to simultaneously track the progress of optimally-thinking LLMs for both performance and efficiency. Second, through comprehensive evaluations of 33 different thinking and non-thinking LLMs, we show that state-of-the-art models struggle to optimally balance accuracy and efficiency, leaving a large gap for improvement in future work. Third, we explore and compare several methods to encourage optimal thinking. Our results show that, while some approaches are promising, there still exists a significant trade-off between efficient and performant LLMs.

## 2 OPTIMAL THINKING BENCHMARK

`OptimalThinkingBench` consists of two complementary benchmarks designed to evaluate the full spectrum of LLMs' thinking behavior. While `OverthinkingBench` measures excessive computation on simple queries, `UnderthinkingBench` quantifies insufficient reasoning on complex tasks. Together, they provide a unified framework for assessing whether models can adaptively balance computational cost with task complexity while maintaining accuracy.

### 2.1 OVERTHINKINGBENCH

`OverthinkingBench` consists of two subsets: `OvT-Math`, focusing on simple math problems and `OvT-General`, consisting of general queries across diverse domains and answer types. In particular, for `OvT-Math`, we use Level 1 and 2 problems from the MATH dataset (Hendrycks et al., 2021). For `OvT-General`, we employ a two-stage pipeline consisting of *Constrained Dataset Generation* followed by *Dataset Filtering*, as illustrated in Figure 5. We follow a fully synthetic dataset creation recipe to ensure that `OverthinkingBench` can also be easily extended and/or difficulty adjusted without human intervention, keeping pace with the rapid progress of LLMs.

**Constrained Dataset Generation.** Creating a benchmark that covers a wide set of queries, in line with real-world query distributions, requires diversity. Naively prompting an LLM would primarily produce degenerate questions that may fail to capture the breadth of user queries (Shypula et al., 2025). To address this, we use a constrained question generation setup: given a pair of constraints $\mathcal{C} = \{D, T\}$ where $D$ represents a specific domain and $T$ an answer type, we prompt an LLM $\mathcal{L}$ to generate $n$ question-answer pairs: $\mathcal{L}(\mathcal{C}) \rightarrow \{(q_i, a_i)\}_{i=1}^n$ where each pair $(q_i, a_i)$ satisfies the specified constraints. We source 72 domains, $D$, that span science (e.g., Mechanics, Quantum Physics), general knowledge (e.g., Global Facts) from SuperGPQA (Du et al., 2025). Our answer types, $T$, include four categories that ensure diverse response formats: (a) numeric answers, (b) multiple-choice questions (MCQ), (c) one-word or short phrase responses, and (d) open-ended answers.

This approach offers several advantages. First, it ensures coverage across domains and answer types. Second, the modular constraints enable systematic ablation studies to understand how overthinking varies with specific domains or answer formats. Third, the generation recipe provides defense against benchmark contamination, since new questions can be generated while maintaining the same properties. In our analysis, we also vary the number of options in MCQs from 4 to 12, allowing us to investigate how distractors affect thinking behavior. The prompt templates are in Appendix C.

**Dataset Filtering.** Synthetically generated benchmarks require validating answer correctness and ensuring both question clarity and appropriate difficulty. Since an LLM generates both questions and answers, filtering becomes essential. Our filtering method takes advantage of the principle that simple questions should elicit consistently correct responses. Thus, for each generated question $q_i$, we sample $k = 8$ responses from a separate LLM $\mathcal{L}'$: $\mathcal{L}'(q_i) \rightarrow \{y_1, y_2, \ldots, y_k\}$.

We retain a question-answer pair $(q_i, a_i)$ if and only if all the sampled answers from the LLM $\mathcal{L}'$ match the answer $a_i$ generated by the LLM $\mathcal{L}$. For answer matching, we use an LLM-as-a-Judge $\mathcal{L}_{\text{judge}}$ that outputs true only if the two answers agree i.e., a data point is accepted if $\forall j \in \{1, \ldots, k\}$ : $\mathcal{L}_{\text{judge}}(q_i, a_i, y_j) = \text{True}$. The exact prompt for $\mathcal{L}_{\text{judge}}$ is presented in Figure 8.

This recipe ensures three properties: (1) Answer Correctness: The agreement among samples validates the reference answer with a high likelihood. (2) Question Clarity: Consistent responses indicate unambiguous phrasing, since ambiguous questions would lead to divergent interpretations and answers. (3) Appropriate Difficulty: The requirement for 100% agreement ensures questions

are simple enough that they don't require extensive reasoning. Questions that pass this filtering constitute the final `OverthinkingBench` dataset.

**Final Statistics.** For `OvT-General`, after filtering, we obtain $n =$1327 high-quality questions, with approximately 330 questions per answer type and about 18 questions per domain. `OvT-Math` consists of 133 questions from Levels 1 and 2 of the MATH dataset.

**Evaluation Metric.** To evaluate models on `OverthinkingBench`, we track both accuracy and the number of thinking tokens generated[2], ensuring that models produce correct answers without excessive computation. First, for accuracy in `OvT-General`, we employ the same LLM-as-a-Judge $\mathcal{L}_{\text{judge}}$ used for dataset filtering to determine the correctness of a model answer $y_i$, for a given question $q_i$ and reference answer $a_i$:

$$\text{Correctness}_i : \mathcal{L}_{\text{judge}}(q_i, a_i, y_i) \rightarrow \{0, 1\}$$

We rely on an LLM for correctness judgment because model responses on `OverthinkingBench` have diverse answer formats that preclude exact matching. For `OvT-Math`, we use mathematical answer matching using the math-verify tool (Kydlíček). Next, using this correctness criterion, we propose Overthinking-Adjusted Accuracy ($\text{OAA}_t$), a unified metric to track a model's accuracy when using fewer than $t$ thinking tokens:

$$\text{OAA}_t = \frac{1}{n} \sum_{i=1}^{n} \left( \text{Correctness}_i \cdot \mathbb{I}(\text{ThinkTokens}_i < t) \right)$$

However, selecting the threshold $t$ presents a challenge, as a small threshold would cause most thinking models to score 0, while a large threshold would not penalize overthinking. Thus, as an aggregated metric, we report the area under the $\text{OAA}_t$ curve, where the x-axis represents the threshold of thinking tokens $t$ and the y-axis represents the corresponding $\text{OAA}_t$ score. The metric is calculated as:

$$\text{AUC}_{\text{OAA}} = \int_0^{t_{\max}} \frac{\text{OAA}_t}{t_{\max}} \, dt \approx \sum_{t=0}^{t_{\max}} \frac{\text{OAA}_t}{t_{\max}} \quad (1)$$

where $t_{\max}$ denotes a pre-defined maximum number of thinking tokens. Our proposed metric has several key properties and advantages: (1) Maximum and minimum values are comparable to accuracy, making it interpretable and easy to measure progress with. (2) Models achieve high scores by simultaneously using minimal tokens (ideally 0) and answering correctly. (3) Both failure cases, where mod-

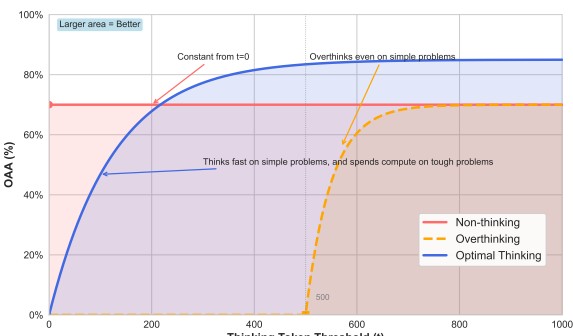

Figure 2: Visualization of $\text{AUC}_{\text{OAA}}$ metric showing Overthinking-Adjusted Accuracy ($\text{OAA}_t$) versus thinking token threshold $t$. We illustrate with three model types: a *Non-thinking* model (red) achieves constant 70% accuracy from t=0; an *Overthinking* model (orange) overthinks even on simple problems, decreasing $\text{AUC}_{\text{OAA}}$; and an *Optimal Thinking* model (blue) thinks fast on simple problems while spending more compute on harder problems, achieving better $\text{AUC}_{\text{OAA}}$. Shaded areas represent $\text{AUC}_{\text{OAA}}$ values. The ranking: $\text{AUC}_{\text{OAA}}^{\text{optimal}} > \text{AUC}_{\text{OAA}}^{\text{non-think}} > \text{AUC}_{\text{OAA}}^{\text{overthink}}$.

els either do not think but generate incorrect answers or generate correct answers but think a lot, will obtain low scores. (4) Despite the integral form, the metric is easily computable since token counts are fixed for each response, reducing the equation to a single term rather than integration. Figure 2 provides a visual illustration of $\text{AUC}_{\text{OAA}}$.

## 2.2 UNDERTHINKINGBENCH

`UnderthinkingBench` is constructed based on a core principle that no matter how large a non-thinking model is, its performance on complex reasoning tasks will be lower than that of a much

---

[2]We count tokens explicitly marked for thinking for each model (e.g., tokens between < think > tags).

smaller thinking model. In other words, it evaluates how necessary "thinking" (via chain-of-thought token generation) is to solve a problem.

**Dataset Generation.** To operationalize this principle, similar to `OverthinkingBench`, we again consider different domains of reasoning problems, moving beyond just math. In particular, we start with 100 different reasoning tasks from Reasoning Gym (Stojanovski et al., 2025) and 4 standard math benchmarks. We evaluate the performance of a small thinking model $P_{small}^{think}$ and a large non-thinking model $P_{large}^{non-think}$ for each task. We retain only tasks where $P_{small}^{think} - P_{large}^{non-think} > \lambda$, with threshold $\lambda = 0.1$. This selection criterion yields: (1) `UT-Reasoning` – a collection of 11 reasoning task types across 6 categories: games, algorithms, graphs, arithmetic, geometry, and logic and (2) `UT-Math`, consisting of competition-level math benchmarks AIME'25 (Art of Problem Solving, 2025) and HMMT'25 (Balunović et al., 2025). Table 5 in the Appendix presents the complete list of tasks from `UT-Reasoning`. For each of these tasks, we procedurally generate questions, allowing us to track progress of underthinking in two model types: (1) Non-thinking models may achieve low accuracy because they cannot generate sufficiently long and correct CoTs. (2) Thinking models may rely on heuristics and underthink on the problems, leading to incorrect answers. The procedural generation enables creation of new questions with increasing complexity to prevent benchmark contamination and to keep up with improving model capabilities.

**Final Statistics.** We generate 550 questions for `UT-Reasoning`, with 50 questions for each of the 11 types of reasoning tasks. `UT-Math` consists 60 questions from AIME'25 and HMMT'25 exams.

**Evaluation Metric.** `UnderthinkingBench` tests the model's ability to generate correct answers to complex reasoning tasks without constraining thinking tokens. We use the task-specific programmatic verifiers provided by Reasoning Gym. In particular, for each sample, we extract the model's final answer from the last `\\boxed{}` in its output and pass it to the task's verifier, which checks correctness against the problem instance via code execution. For example, in the maze shortest-path task, the verifier simulates the proposed path to check for its validity and compares its length to an algorithmically computed optimal solution. For `UT-Math` we use answer match based on the math-verify tool (Kydlíček). Our final score is the macro average across the reasoning and math subsets.

## 2.3 EVALUATION METRIC OF OPTIMALTHIKINGBENCH

The goal of `OptimalThinkingBench` is to track progress through a single unified metric, since overthinking and underthinking are two sides of the same problem. To standardize evaluation across both benchmarks, we combine $AUC_{OAA}$ from `OverthinkingBench` and accuracy $Acc_{ut}$ from `UnderthinkingBench` into a single $F_1$ score: $F_1^{otb} = 2 \cdot \frac{AUC_{OAA} \cdot Acc_{ut}}{AUC_{OAA} + Acc_{ut}}$ Overall, a model scoring high on `OptimalThinkingBench` must avoid overthinking on simple problems and underthinking on complex ones. This metric ensures that models must perform well on both benchmarks simultaneously to achieve high scores, as $F_1$ tends to be closer to the lower of the two component metrics.

## 3 EXPERIMENTS

### 3.1 EXPERIMENTAL SETUP

For generating questions ($\mathcal{L}$), filtering ($\mathcal{L}'$), and evaluation ($\mathcal{L}_{judge}$), we use the same LLM: Llama-4-Maverick with different prompts listed in Appendix C. For evaluation, we set the maximum number of thinking tokens $t_{max} = 1000$ in Equation 1. We create `UnderthinkingBench` using Qwen3-1.7B as the thinking model and Qwen3-235B-A22B as the non-thinking model, and compute performance using similar method as described in the Evaluation section. For each task/dataset, we evaluate 50 questions on 8 random seeds and take the average. We use $\lambda = 0.1$ to obtain a representative set of tasks satisfying the principle that "for certain questions, no matter how large a non-thinking model is, its performance on complex reasoning tasks will be lower than that of a much smaller yet thinking model," while also accounting for statistical uncertainty in estimating model performance. We evaluate 33 different open-source and proprietary models with varying model sizes and different families. For hybrid models, we evaluate them in both thinking and non-thinking modes. We compare models on the complete `OptimalThinkingBench` using $F_1^{otb}$ metric. Full details are in Appendix B.

Table 1: Main results on `OptimalThinkingBench` comparing open/closed thinking/non-thinking models based on accuracy, thinking tokens, and our proposed metrics. The main metrics for over, under, and optimal-thinking are $\text{AUC}_{\text{OAA}}$, accuracy, and $F_1^{\text{otb}}$ respectively. These metrics are bolded for the best performing model in each of the four categories. $^\dagger$ = Hybrid models evaluated in either thinking or non-thinking mode. *Only thinking tokens are counted.

| Model | OptimalThinkingBench $F_1^{\text{otb}}$ ↑ | OverthinkingBench | | | UnderthinkingBench | |
|---|---|---|---|---|---|---|
| | | Accuracy (%) ↑ | Tokens* ↓ | $\text{AUC}_{\text{OAA}}$ ↑ | Accuracy (%) ↑ | Tokens ↓ |
| *Open Non-Thinking Models* | | | | | | |
| Llama-4-Scout | 19.1 | 95.0 | 0 | 95.0 | 10.6 | 904 |
| Llama-4-Maverick | 27.9 | 95.7 | 0 | 95.7 | 16.3 | 993 |
| Qwen2.5-7B | 9.6 | 93.6 | 0 | 93.6 | 5.1 | 1370 |
| Qwen2.5-72B | 19.0 | 96.3 | 0 | 96.3 | 10.5 | 1174 |
| Qwen3-1.7B$^\dagger$ | 12.9 | 89.0 | 0 | 88.8 | 6.9 | 1943 |
| Qwen3-8B$^\dagger$ | 24.5 | 95.9 | 0 | 95.8 | 14.0 | 2223 |
| Qwen3-235B-A22B$^\dagger$ | **31.7** | 96.9 | 0 | **96.7** | **18.9** | 1501 |
| *Closed Non-Thinking Models* | | | | | | |
| Sonnet-4$^\dagger$ | **48.3** | **97.4** | 0 | **97.4** | **32.1** | 2229 |
| GPT-4o | 17.8 | 95.3 | 0 | 95.3 | 9.8 | 694 |
| GPT-4.1 | 35.4 | 97.1 | 0 | 97.1 | 21.7 | 1846 |
| *Open Thinking Models* | | | | | | |
| Magistral-Small-2506 | 11.2 | 95.7 | 3303 | 6.4 | 42.9 | 16788 |
| R1-Distill-Llama-8B | 20.7 | 93.2 | 1307 | 21.7 | 19.8 | 11113 |
| Qwen3-1.7B$^\dagger$ | 24.2 | 93.8 | 1519 | 20.6 | 29.2 | 13072 |
| Qwen3-8B$^\dagger$ | 24.3 | 98.1 | 1588 | 16.3 | 47.7 | 13858 |
| R1-0528-Qwen3-8B | 28.8 | 96.6 | 1926 | 24.2 | 35.7 | 15610 |
| Qwen3-235B-A22B$^\dagger$ | 23.2 | 98.3 | 1632 | 14.6 | 55.5 | 12057 |
| GPT-OSS-20B | 57.3 | 97.1 | 467 | 72.7 | 47.3 | 8937 |
| GPT-OSS-120B | **68.3** | 97.1 | 154 | **83.3** | 57.9 | 4968 |
| *Closed Thinking Models* | | | | | | |
| Sonnet-4$^\dagger$ | 64.2 | 99.3 | 706 | 71.3 | 58.3 | 14035 |
| o3 | **71.1** | 97.5 | 235 | **78.6** | **65.0** | 6273 |

## 3.2 MAIN RESULTS WITH THINKING AND NON-THINKING MODELS

In Table 1, we show the performance of 20 representative models on `OptimalThinkingBench` with full results for all 33 models in Table 6 (Appendix D). Our evaluation reveals the following key findings on the state of current thinking and non-thinking LLMs.

**Models fail to achieve optimal balance between accuracy and efficiency.** Comparing our primary $F_1^{\text{otb}}$ metric, we observe that o3 achieves the best performance on `OptimalThinkingBench` at 71.1%. Among the open-weight models, the best results are obtained by the GPT-OSS-120B model at 68.3%, representing a 3-point gap compared to the best closed-weight model. Apart from GPT-OSS, all other open-weight models score below 50% on our benchmark. Overall, no current model effectively balances efficiency and reasoning capability because they either do well on `OverthinkingBench` or `UnderthinkingBench` but not on both at the same time. This gap demonstrates substantial room for improvement in developing *unified* models (particularly with open recipes and weights) that can adaptively adjust their computational effort based on task complexity.

**Most thinking models exhibit severe overthinking on simple queries.** On `OverthinkingBench`, all thinking models generate at least 100 thinking tokens for simple queries, with most models generating more than 1300 tokens. This is reflected in the $\text{AUC}_{\text{OAA}}$ scores that are much lower than the corresponding raw accuracy numbers. The best-performing open and closed thinking models are GPT-OSS-120B and o3, generating 154 and 235 tokens respectively. However, other models such as Qwen3 utilize between 1373-1632 tokens, while Magistral utilizes over 3300 tokens. Since most queries in this benchmark are simple questions such as "If a steel rod is 1 meter long, what is its length in centimeters?", the unnecessary computation severely penalizes their $\text{AUC}_{\text{OAA}}$ scores, highlighting increased cost and reduced utility for users. In contrast, non-thinking models achieve much higher $\text{AUC}_{\text{OAA}}$ scores, matching their raw accuracies.

**Thinking models, however, show substantial gains on complex reasoning.** Despite overthinking on simple queries, thinking models are much better than non-thinking models on `UnderthinkingBench`. o3 obtains the highest accuracy on `UnderthinkingBench` at

Table 2: Results comparing different methods for improving optimal thinking on our benchmark. We evaluate on both `OverthinkingBench` and `UnderthinkingBench` to understand how methods developed to reduce overthinking impact underthinking.

| Method | OptimalThinkingBench $F_1^{\text{otb}}$ ↑ | OverthinkingBench | | | UnderthinkingBench | |
|---|---|---|---|---|---|---|
| | | Accuracy (%) ↑ | Tokens ↓ | AUC$_{\text{OAA}}$ ↑ | Accuracy (%) ↑ | Tokens ↓ |
| R1-Distill-Qwen-7B | 24.5 | 91.5 | 1172 | 25.4 | 23.6 | 11763 |
| + VeriThinker (Chen et al., 2025b) | 27.4 (+2.9) | 91.9 (+0.4) | 689 (-41%) | 46.2 (+20.8) | 19.4 (-4.2) | 5954 (-49%) |
| + SB-DS | 24.3 (-0.2) | 82.2 (-9.3) | 110 (-91%) | 73.9 (+48.5) | 14.5 (-9.1) | 3155 (-73%) |
| + L1 (Aggarwal & Welleck, 2025) | 20.8 (-3.7) | 91.1 (-0.4) | 1037 (-12%) | 19.9 (-5.5) | 21.8 (-1.8) | 2853 (-76%) |
| + AdaptThink (Zhang et al., 2025a) | 38.3 (+13.8) | 90.8 (-0.7) | 211 (-82%) | 77.2 (+51.8) | 25.4 (+1.8) | 10224 (-13%) |
| Qwen3-8B | 24.3 | 98.1 | 1588 | 16.3 | 47.7 | 13858 |
| + Model Merging (Wu et al., 2025) | 38.2 (+13.9) | 97.6 (-0.5) | 1024 (-36%) | 32.4 (+16.1) | 46.5 (-1.2) | 11738 (-15%) |
| + L1 (Aggarwal & Welleck, 2025) | 28.5 (+4.2) | 97.5 (-0.6) | 867 (-45%) | 24.2 (+7.9) | 34.6 (-13.1) | 4867 (-65%) |

65.0%, followed by GPT-OSS-120B at 57.9%. Analyzing models that operate in hybrid mode, all Qwen3 models score at or below 20% accuracy in non-thinking mode, with Qwen3-32B achieving only 14.9%. However, when these same models operate in thinking mode, their performance increases significantly. For example, Qwen3-14B's accuracy in thinking mode increases from 14% to 52.4%, representing a 38.4% improvement. This pattern also holds for other hybrid models.

## 3.3 METHODS FOR IMPROVING OPTIMAL THINKING

Given that all models exhibit a trade-off between performance and efficiency, we now explore different approaches to encourage optimal thinking in models. These include: (1) methods for efficient reasoning that mitigate overthinking, (2) routing between thinking and non-thinking modes based on the question difficulty, and (3) explicitly prompting models to not overthink or underthink.

**Efficient reasoning methods reduce overthinking but also affect performance.** Our first approach toward improving optimal thinking is to mitigate overthinking in thinking models using recently proposed methods for efficient reasoning. We test five such methods implemented with two kinds of thinking models, as shown in Table 2.[3] They are based on the following concepts **1. Length-based Reward Shaping:** L1 and AdaptThink (Aggarwal & Welleck, 2025; Zhang et al., 2025a) primarily modify the reward function during RL training to include an additional length term along with original correctness reward. **2. Model Merging:** This method (Wu et al., 2025) merges weights of two different models with different output length distributions, to enable short CoT on simple and long CoT on complex questions. **3. Auxiliary Task Training:** VeriThinker (Chen et al., 2025b) shows that training for verification task leads to more efficient reasoning.

Generally, these methods reduce token usage in the range of 12% to 91% on `OverthinkingBench`. However, on `UnderthinkingBench`, there is a clear decrease in accuracy (of up to 13%) in 5 out of these 6 model-method combinations compared to their base versions,. This results in 2 out of these 6 configurations underperforming on the overall $F_1^{\text{otb}}$ score, indicating that efficiency gains come at the cost of reasoning capability. The only exception is AdaptThink that improves on `UnderthinkingBench` and significantly reduces thinking tokens on `OverthinkingBench`, although at the expense of a small drop in accuracy (0.7%). For example, R1-Distill-Qwen-7B achieves 24.5 $F_1^{\text{otb}}$ score, but with L1, this drops to 20.8% despite some token reduction (from 1172 to 1037 tokens on `OverthinkingBench`).

Recall that `OptimalThinkingBench` contains non-math subsets of data and interestingly, we find that even when there are efficiency gains from these methods, they are often less pronounced for non-math tasks (e.g., 37% instead of 82% with AdaptThink), highlighting that training for efficient math reasoning may not always generalize to other domains (Table 7 and Appendix D).

**Question-difficulty based routing helps optimality but still has a large gap to the oracle router.** Our next approach is to leverage a router model that uses non-thinking mode for simple questions and thinking mode for complex questions. We evaluate an open-source router (Tran et al., 2025) on Qwen3 models (that support hybrid modes), comparing against both the best individual mode performance and an oracle router that always selects the optimal mode. This trained router is a publicly available state-of-the-art model that is prompted to classify queries as simple or complex, using

---

[3]We directly evaluate these models from HuggingFace, without retraining them.

Table 3: Comparison of a state-of-the-art router (that routes between non-thinking and thinking modes based on question difficulty) with an oracle router on Qwen3 to encourage optimal thinking.

| Method | OptimalThinkingBench $F_1^{\mathbf{otb}}$ ↑ | OverthinkingBench | | | UnderthinkingBench | |
|---|---|---|---|---|---|---|
| | | Accuracy (%) ↑ | Tokens ↓ | AUC$_{\mathbf{OAA}}$ ↑ | Accuracy (%) ↑ | Tokens ↓ |
| Qwen3 (Avg) | 24.3 | 97.0 | 1544 | 17.1 | 45.9 | 24074 |
| Qwen3-NonThink (Avg) | 23.7 | 94.5 | 0 | 94.5 | 13.7 | 3545 |
| w/ Trained Router | 46.9 (+20.4%) | 95.9 | 876 | 55.2 | 41.7 | 22238 |
| w/ Oracle Router | 61.2 | 94.5 | 0 | 94.5 | 45.9 | 24074 |

Table 4: Results comparing different prompt variations to encourage optimal thinking.

| Method | OptimalThinkingBench $F_1^{\mathbf{otb}}$ ↑ | OverthinkingBench | | | UnderthinkingBench | |
|---|---|---|---|---|---|---|
| | | Accuracy (%) ↑ | Tokens ↓ | AUC$_{\mathbf{OAA}}$ ↑ | Accuracy (%) ↑ | Tokens ↓ |
| Standard | 26.3 | 96.7 | 1493 | 19.4 | 43.1 | 13207 |
| Step-by-Step | 18.3 (-8.0) | 96.4 (-0.3) | 1638 (+10%) | 12.0 (-7.4) | 43.2 (+0.1) | 13580 (+3%) |
| Don't Overthink | 34.0 (+7.7) | 96.7 (+0.0) | 1147 (-23%) | 29.5 (+10.1) | 42.0 (-1.1) | 12424 (-6%) |

which we route to non-thinking or thinking mode, respectively. We also evaluate against an oracle router that chooses the non-thinking mode for `OverthinkingBench` and the thinking mode for `UnderthinkingBench`. Table 3 shows the results aggregated across different Qwen3 models (full results in Appendix D). While the model-based router improves upon the best individual mode by 20.4%, it still falls significantly short of the result obtained by an oracle router, with gap ranging around 15%. These results suggest that while routing techniques may provide benefits in specific scenarios, developing effective routers for general unified reasoning remains an open challenge.

**Explicitly Prompting Models.** Next, we explore whether models can be explicitly prompted to think optimally. In particular, we use the following prompt suffixes: **1.) Don't Overthink**: We explicitly prompt models to not overthink. **2.) Let's think step-by-step**: This is a standard prompt suffix, often used in real-world queries to encourage models to think step-by-step (Kojima et al., 2022). Table 4 shows the results of different prompt variations on `OverthinkingBench` macro-averaged across Qwen3 models (with full results in Table 9). First, encouragingly on `OverthinkingBench`, we find that prompting models to not overthink leads to a consistent drop in tokens used (on average by 23%), without impacting accuracy. In contrast, prompting models to "Think step-by-step" leads to a small drop in accuracy across all Qwen models, while importantly, increasing thinking length by roughly 10%. This suggests that the colloquial prompt suffix further aggravates overthinking for simple queries with thinking models. Overall, these results point to the fact that the amount of thinking in LLMs can vary noticeably based on the exact prompts being used. We hope that our benchmark generally encourages further explorations of all these training-time and inference-time strategies of optimal thinking in future models.

## 3.4 ANALYSIS OF OVERTHINKING AND UNDERTHINKING

**Analysis of Math vs General Domain.** Figure 3 compares different metrics on `OvT-Math` and `OvT-General` with 4 representative models. Interestingly, while models achieve higher accuracy on `OvT-Math` than in `OvT-General`, the latter also results in much higher overthinking. This result is specifically striking for GPT-OSS-20B which thinks ten times more on Math, even though these questions are relatively simple. This shows that optimizing overthinking for a specific domain may not generalize to other domains and hence may not enhance overall optimal thinking in LLMs.

**Analysis by Question Domain and Answer Types.** We analyze how overthinking varies across different question characteristics for `OvT-General` (full analysis in Appendix D.2). Figure 4a shows thinking token usage across domains sorted by average thinking length. Models generate significantly more tokens for STEM domains (Engineering, Economics) compared to History, yet this increased thinking shows no correlation with accuracy (Spearman $\rho = -0.46$, $p > 0.05$) or performance improvements over non-thinking counterparts ($\rho = 0.29$, $p > 0.05$), demonstrating models cannot adaptively adjust thinking based on domain complexity. Moreover, Figure 4b shows how answer types affect thinking: while models use comparable tokens for MCQ and open-ended questions, they consume substantially more for numeric questions. Crucially, unlike `OvT-Math`, these numeric questions are often simple facts, yet they trigger extensive overthinking without ac-

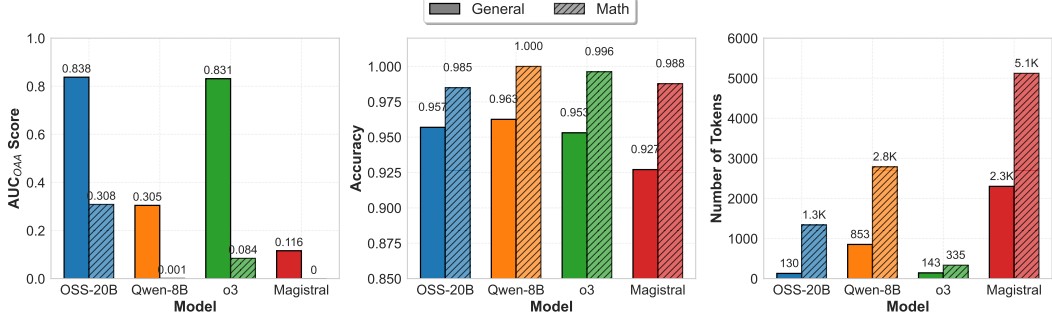

Figure 3: Comparison of overthinking metrics on `OvT-Math` and `OvT-General`. Math questions invoke greater overthinking than general-domain ones.

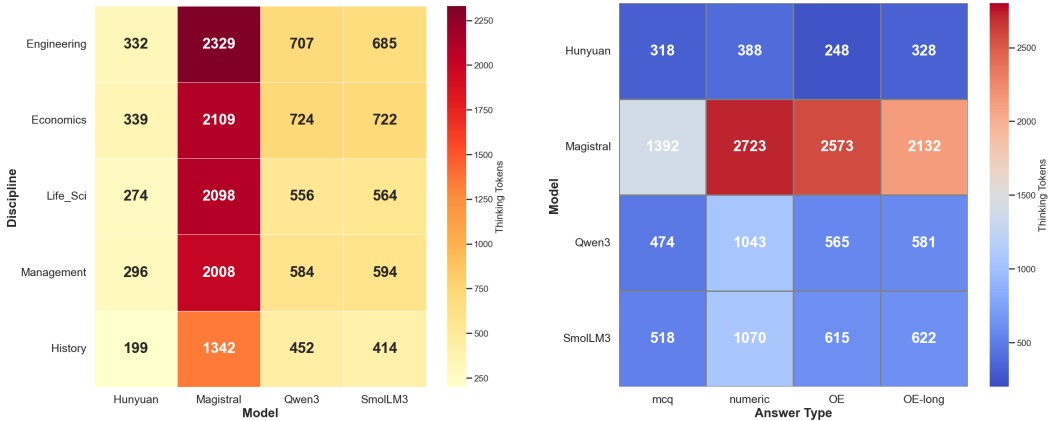

(a) Thinking tokens across problem domains.

(b) Thinking tokens across answer types.

curacy benefits in 4/5 models tested. This is likely because of the emphasis of post-training on mathematical tasks that causes models to conflate numerical tokens with computational complexity. Additionally, Figure 10 demonstrates that adding completely irrelevant MCQ distractors in questions causes near-linear increase in overthinking (42 tokens per option, $R^2 = 0.94$). These patterns reveal that current models may be relying on superficial cues (domain keywords, numerical tokens, option count) rather than actual task complexity when allocating computational resources.

**Qualitative Analysis.** We qualitatively examine failure modes in both sub-benchmarks using statistically significant examples obtained by generating 128 responses per model and selecting cases where performance differences are robust (Full details in Appendix D.4). Examples 1, 2, and 3 demonstrate a recurring overthinking pattern: models initially identify correct answers but subsequently overthink, introducing conflicting information or flawed reasoning that leads to incorrect conclusions. Conversely, Examples 4 and 5 reveal underthinking behavior where non-thinking models rely on heuristics without verification—claiming to use algorithms like BFS while actually taking the first plausible path without systematic exploration or validation. These patterns illustrate how overthinking creates unnecessary confusion while underthinking omits essential verification steps.

## 4    RELATED WORK

Recent works have analyzed overthinking and underthinking in LLMs across various domains including adversarial, tool-use, math, and unanswerable queries (Sui et al., 2025; Wang et al., 2025b; Chen et al., 2025a; Kumar et al., 2025; Cuadron et al., 2025; Song & Zheng, 2025; Zhao et al., 2025; Kirichenko et al., 2025; Liu et al., 2025; TSB, 2025). However, these studies treat overthinking and underthinking in isolation on specific benchmarks. Further, existing works on efficient reasoning primarily target overthinking through RL-based length penalties (Aggarwal & Welleck, 2025; Arora & Zanette, 2025; Yi et al., 2025; Zhang et al., 2025a), verification training (Chen et al., 2025b), early exit strategies (Yang et al., 2025; Jiang et al., 2025), or inference-time interventions (Wang

et al., 2025a). Others address underthinking by forcing longer generation through decoding time interventions (Muennighoff et al., 2025; Jin et al., 2025). However, these typically rely on disparate evaluation setups and use their own unique metrics to measure overthinking or underthinking, making fair comparison across approaches difficult and hindering systematic progress. In contrast, `OptimalThinkingBench` provides the first unified benchmark with standardized metrics for both overthinking and underthinking, making evaluation more standardized and enabling fair comparison between these methods. It spans several general and reasoning domains (including math) and reveals that optimizing for one of over and underthinking typically degrades the other. We refer readers to Appendix A for a more detailed related work.

## 5 CONCLUSION

We proposed `OptimalThinkingBench`, a new benchmark to jointly measure overthinking and underthinking in LLMs. Our benchmark consists of two sub-benchmarks, spanning math and general-domain questions in 72 domains, with four answer types, and belonging to diverse reasoning tasks. Through a combined efficiency-adjusted accuracy metric and multiple sub-metrics, we evaluated 33 state-of-the-art thinking and non-thinking models and showed that no model is able to optimally balance performance and efficiency on our benchmark. We also explored different methods to encourage such optimal thinking which only rarely resulted in improvements, highlighting the need for better unified and optimally-thinking LLMs in the future. `OptimalThinkingBench` is designed to evolve with increasing model competence, providing a tunable method to benchmark the optimal thinking performance of new models.

## REPRODUCIBILITY STATEMENT

We have uploaded `OptimalThinkingBench` as part of the supplementary material and it will be made public upon acceptance. In Appendix C, we also show our prompts to help reproduce and potentially extend our benchmark in the future. Our evaluation code will also be released to enable benchmarking and measuring progress on `OptimalThinkingBench`.

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

## A    DETAILED RELATED WORK

**Overthinking and Underthinking in LLMs.** Several recent works have analyzed the issues of both overthinking and underthinking in LLMs (Sui et al., 2025; Wang et al., 2025b; Chen et al., 2025a; Saha et al., 2024; Zhang et al., 2025b; Pu et al., 2025). Notably, these analyses span adversarial (Kumar et al., 2025), tool-use (Cuadron et al., 2025), math (Song & Zheng, 2025; Zhao et al., 2025; Su et al., 2025; Wang et al., 2025b) and unanswerable (Kirichenko et al., 2025) queries. Furthermore, Liu et al. (2025) show that chain-of-thought can hurt performance in tasks where deliberation hurts performance in humans. Additionally, a very recent concurrent blog post introduces a benchmark and discusses the problem of token efficiency in thinking models (TSB, 2025). Many of these studies have treated overthinking and underthinking in isolation, without unified metrics, often on different and specialized benchmarks, which has hindered the ability to effectively track progress toward optimal thinking in LLMs. OptimalThinkingBench addresses this issue by providing a unified benchmark and metrics, thereby demonstrating that independently optimizing models for overthinking or underthinking results in improvements in only one of these at the expense of the other.

**Methods for Addressing Overthinking and Underthinking.** A large body of prior work has explored reducing overthinking in models with efficient reasoning methods (Arora & Zanette, 2025; Kang et al., 2024; Fang et al., 2025). For instance, Aggarwal & Welleck (2025); Arora & Zanette (2025); Yi et al. (2025); Zhang et al. (2025a) modify reinforcement learning objectives, VeriThinker (Chen et al., 2025b) trains models on verification tasks, Yang et al. (2025); Jiang et al. (2025) develop early exit methods, and Wang et al. (2025a) propose a simple inference time intervention. However, these methods have almost universally focused on math and code domains, neglecting the vast proportion of general user queries (Handa et al., 2025). Similarly, past works have improved underthinking by forcefully adding tokens when the model is about to stop generation (Muennighoff et al., 2025; Jin et al., 2025). Furthermore, they typically rely on disparate evaluation setups and use their own unique metrics to measure overthinking or underthinking, making fair comparison across approaches difficult and hindering systematic progress. `OptimalThinkingBench` addresses this gap by providing a unified interface (with benchmarks and metrics) to study both overthinking and underthinking. This makes evaluation more standardized and enables fair comparison between these methods. Using this evaluation setup, we compare several of these past methods to show that while existing efficient reasoning methods improve overthinking, they often also degrade underthinking.

## B    EXPERIMENTAL SETUP

### B.1    OPTIMALTHINKINGBENCH CREATION

For generating questions ($\mathcal{L}$), filtering ($\mathcal{L}'$), and evaluation ($\mathcal{L}_{\text{judge}}$), we use the same LLM: Llama-4-Maverick with different prompts listed in Appendix C. For `OverthinkingBench`, we use 72 different domains, 4 different answer types, and for each (domain, answer type) pair, we generate a maximum of 5 questions. For filtering, we sample 8 responses for each question. We use temperature = 0.6 and top_p = 1.0. For evaluation, we set the maximum number of thinking tokens $t_{\max} = 1000$ in Equation 1. In creating `UnderthinkingBench`, we set the threshold $\lambda = 0.1$ and use Qwen3-1.7B as the thinking model and Qwen3-235B-A22B as the non-thinking model.

### B.2    MODEL EVALUATION

We evaluate 33 different open-source and proprietary models on `OptimalThinkingBench`, with varying model sizes, and different families. For hybrid models, we evaluate them in both thinking and non-thinking modes. We compare models on the complete `OptimalThinkingBench` based on our $F_1^{\text{otb}}$ metric. In addition, for each model, we report the number of thinking tokens, accuracy, and $\text{AUC}_{\text{OAA}}$ for `OverthinkingBench`, and accuracy, complete output tokens for `UnderthinkingBench`. We report complete output tokens, because answers for `UnderthinkingBench` are typically only a few tokens, and it is well-studied that even chain-of-thought tokens outside of thinking tags contribute to higher performance. All evaluations are performed over 8 seeds, and consistent temperature sampling of 0.6.

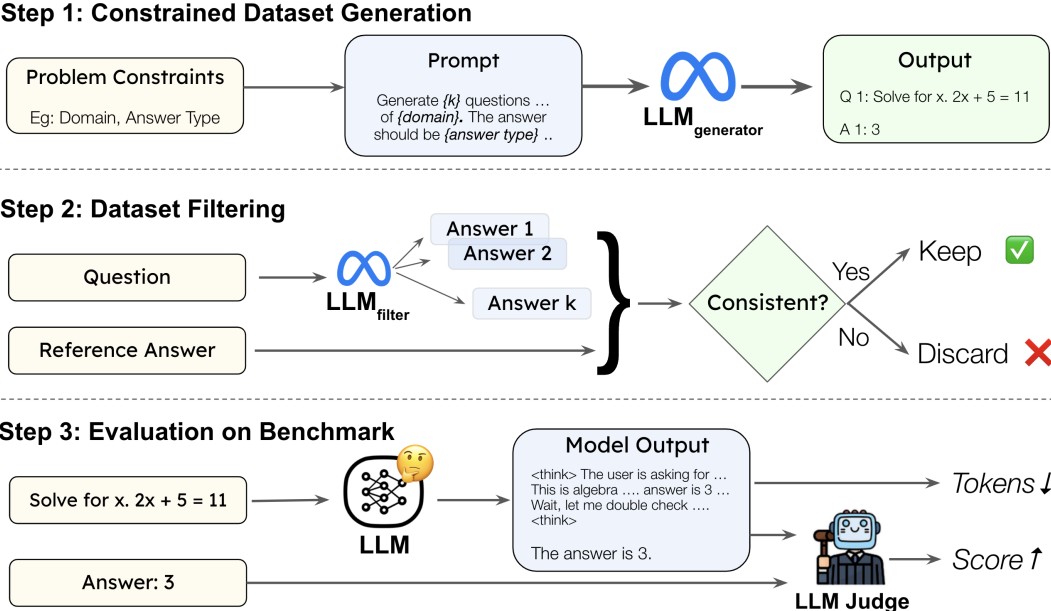

Figure 5: Generation recipe of `OverthinkingBench` (Step 1 and 2) and evaluation recipe of models on `OverthinkingBench` (Step 3). We follow a generation and filtering pipeline to generate and verify the questions and answer correctness. We evaluate model outputs on this benchmark based on the number of tokens used (overthinking) and answer correctness, using an LLM-as-a-Judge verifier.

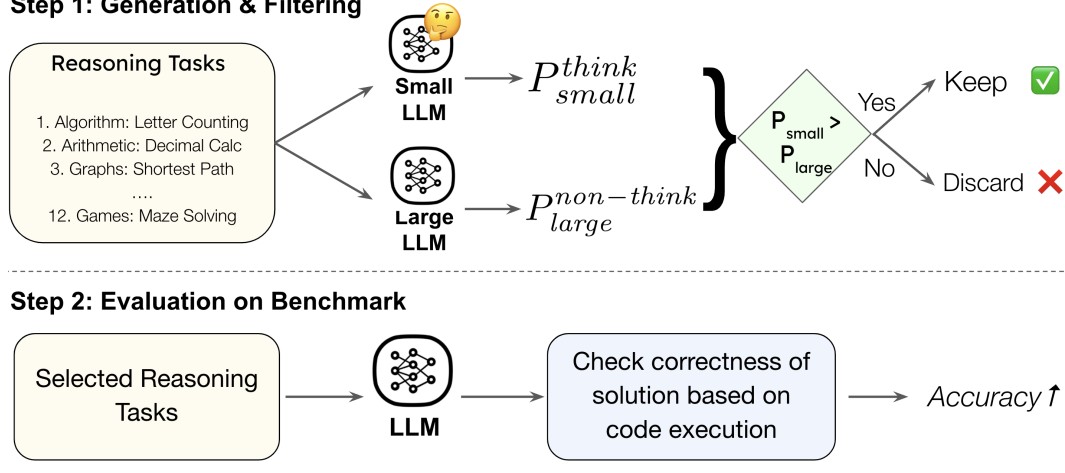

Figure 6: Generation recipe of `UnderthinkingBench` (Step 1) and evaluation recipe of models on `UnderthinkingBench` (Step 2). We follow a generation and filtering pipeline to first generate and then check for reasoning tasks that particularly benefit from thinking (by leveraging the difference between a small thinking model and a large non-thinking model). We evaluate models on `UnderthinkingBench` using accuracy computed with a code-based verifier.

## C PROMPTS AND ADDITIONAL DETAILS OF OPTIMALTHINKINGBENCH

This section consists of all prompts used throughout `OverthinkingBench` for data generation, filtering, and evaluation.

> **Main prompt for generating `OverthinkingBench` questions**
>
> ```
> Suppose I have this problem.  So basically there are these recent
> models that are called reasoning models, and the idea is that if you
> increase the inference compute, in the sense that if they generate
> longer chain of thoughts, the accuracy increases.  However, one big
> challenge with them is that they sometimes overthink, spending a
> lot of compute even on simple questions, results in reduced utility
> for user, as it takes a lot of time generating long thinking.  A
> simple question could be 2+2, and the model is expected to answer
> immediately.
>
> In order to evaluate this behavior I plan to propose
> OverthinkingBench.  The idea is simple, this benchmark would contain
> some very simple questions, where model is not expected to think for
> more than 10 to 20 tokens, and sometimes 0 tokens to answer them.
> The accuracy would mostly be 100% because the questions would be
> simple, and our evaluation would be average tokens used, standard
> deviation, or thinking violations (how many times thinking was > 20
> tokens).
>
> Now I want you to make prompts for such a dataset.  Note that
> distribution of prompts should be similar to standard benchmarks
> for Large language models.  Simple questions, although of varying
> difficulty, varying domains, varying types.  Your goal is to create
> 50 such prompts.  Diversity along different dimensions is expected.
>
> OUTPUT FORMAT: output json, List[dict], where dict contains two
> keys:  "Question", "Answer category"
>
> Domains:  I will give you the following domains, and you are
> expected to generate some simple (not at all tough) questions.
> This is to ensure the benchmark contains real world queries.  The
> questions can be straightforward factual questions, require some
> very basic multi-hop reasoning, or some very basic math questions.
>
> {question_format}
>
> Here are the {len(domains)} domains:
> {domains}
>
> For each domain create 5 questions.
> ```

Figure 7: Main prompt for constructing the `OvT-General` subset of `OverthinkingBench`.

## C.1 QUESTION GENERATION

The core prompt for generating simple and general questions across diverse domains and answer types is shown in Figure 7. This prompt is designed to elicit questions that should require minimal reasoning tokens while maintaining diversity across domains and answer formats.

## C.2 ANSWER FORMAT SPECIFICATIONS

`OverthinkingBench` supports four distinct answer types, each with specific constraints to ensure sound evaluation. The format specifications are provided as template substitutions in the main generation prompt.

---

**Prompt specification for numeric answer questions**

```
Additionally answer to every question should be a numeric value,
that can be matched to gold answer.  However, the answer to the
question should be clear.  Answer in similar json format.

## Important:  Make sure questions have 1 clear numerical answer
with no ambiguity or any potential similar or nearby answer.

For example, a bad question would be:  ``How many people are
affected by diabetes worldwide in millions?''  This is a bad
question, because the number keeps on changing every year, is based
on estimate and therefore answers could vary.  Do not output such
questions.
```

---

**Prompt specification for multiple choice questions**

```
Additionally every question will be MCQ, with only one correct
option and total of {num_options} options, of which clearly 1 is
correct without ambiguity.  Answer in similar json format.
```

---

**Prompt specification for short answer questions**

```
Additionally answer to every question should be a short answer
such as single word or phrase, that can be matched to gold answer.
However, the answer to the question should be clear.  Answer in
similar json format.
```

---

**Prompt specification for long answer questions**

```
Additionally answer to every question should be a long answer such
as a paragraph, that will be judged by a separate LLM as judge
against the reference answer.  However, the answer to the question
should be clear without ambiguity.  Answer in similar json format.
```

---

## C.3 LLM-AS-A-JUDGE VERIFICATION

Once the model generates the answer, we first extract the answer within the last `\\boxed{}` in its output, using a regular expression. The answer is then passed to the LLM-as-a-Judge. If `\\boxed{}` is not found, we provide the complete output (excluding the reasoning trace) to the LLM-as-a-Judge. The verification prompt for the LLM-as-a-Judge is shown in Figure 8.

---

**LLM-as-a-Judge prompt for answer verification**

```
User:  ### Question:  {question}

### Ground Truth Answer:  {ground_truth}

### Student Answer:  {student_answer}

For the above question, please verify if the student's answer is
equivalent to the ground truth answer.

Do not solve the question by yourself; just check if the student's
answer is equivalent to the ground truth answer.

If the student's answer is correct, output ''Final Decision:  Yes''.
If the student's answer is incorrect, output ''Final Decision:
No''.

Assistant:
```

Figure 8: LLM-as-a-Judge Answer Verification Prompt.

## C.4   DOMAIN COVERAGE

`OverthinkingBench` spans 72 distinct domains to ensure comprehensive coverage of real-world query distributions. These domains are sourced from SuperGPQA and are shown in Figure 9.

---

**Complete list of 72 domains used in `OverthinkingBench` Creation**

```
Electronic Science and Technology, Philosophy, Traditional
Chinese Medicine, Applied Economics, Mathematics, Physics,
Clinical Medicine, Computer Science and Technology, Information
and Communication Engineering, Control Science and Engineering,
Theoretical Economics, Law, History, Basic Medicine, Education,
Materials Science and Engineering, Electrical Engineering,
Systems Science, Power Engineering and Engineering Thermophysics,
Military Science, Biology, Business Administration, Language
and Literature, Public Health and Preventive Medicine, Political
Science, Chemistry, Hydraulic Engineering, Chemical Engineering and
Technology, Pharmacy, Geography, Art Studies, Architecture, Forestry
Engineering, Public Administration, Oceanography, Journalism and
Communication, Nuclear Science and Technology, Weapon Science and
Technology, Naval Architecture and Ocean Engineering, Environmental
Science and Engineering, Transportation Engineering, Geology,
Physical Oceanography, Musicology, Stomatology, Aquaculture,
Mechanical Engineering, Aeronautical and Astronautical Science
and Technology, Civil Engineering, Mechanics, Petroleum and
Natural Gas Engineering, Sociology, Food Science and Engineering,
Agricultural Engineering, Surveying and Mapping Science and
Technology, Metallurgical Engineering, Library Information and
Archival Management, Mining Engineering, Astronomy, Geological
Resources and Geological Engineering, Atmospheric Science, Optical
Engineering, Animal Husbandry, Geophysics, Crop Science, Management
Science and Engineering, Psychology, Forestry, Textile Science and
Engineering, Veterinary Medicine, Instrument Science and Technology,
Physical Education
```

Figure 9:  List of all domains used in `OverthinkingBench`.

## C.5   ADDITIONAL DETAILS OF UNDERTHINKINGBENCH

`UnderthinkingBench` utilizes existing challenging reasoning tasks from the Reasoning Gym framework.  Rather than using custom prompts, we leverage 11 pre-defined reasoning task types with specific parameter configurations.  Table 5 provides an overview of all tasks along with their categories and descriptions.

Table 5: Reasoning tasks and configurations for underthinking benchmark.

| Reasoning Task | Category | Description |
|---|---|---|
| ab | Algorithmic | Pattern recognition in sequences |
| Letter Counting | Algorithmic | Count specific letters in given text |
| Bitwise Arithmetic | Arithmetic | Execute bitwise operations on binary numbers |
| Fraction Simplification | Arithmetic | Simplify fractions to their lowest terms |
| Quantum Locks | Graphs | Find shortest sequence to reach correct value |
| Maze | Games | Navigate through the maze to reach destination |
| Knight Swap | Games | Swap all positions of black knights with white knights |
| Puzzle 24 | Games | Use four numbers to make 24 with operations |
| Tsumego | Games | Solve Go game tactical problems |
| Advanced Geometry | Geometry | Solve advanced geometry problems |
| Propositional Logic | Logic | Infer correct conclusions from given premises |

Each reasoning task generates 50 instances, resulting in a total of 550 challenging problems that require substantial computational effort to solve correctly. The tasks span six domains: games (maze, knight swap, puzzle 24, tsumego), algorithms (ab, letter counting), graphs (quantum locks), arithmetic (bitwise arithmetic, fraction simplification), geometry (advanced geometry) and logic (propositional logic).

# D  ADDITIONAL RESULTS AND ANALYSES

## D.1  METHODS FOR IMPROVING OPTIMAL THINKING

Full results on 33 models are shown in Table 6.

This section contains the full results for the methods mentioned in the main paper.

Results for efficiency-based methods on the non-math subsets are shown in Table 7. Results for router based methods are in Table 8. Results for prompt based methods are in Table 9.

## D.2  OVERTHINKING ANALYSIS

**Analysis by Answer Types.** In Table 10 we evaluate hybrid models like Qwen3 and compare their accuracy differences between thinking and non-thinking modes across four answer types from our `OverthinkingBench`. Qwen3 allows switching between the two modes through its chat templates. Results where thinking statistically improves performance are marked in green, while statistically significant degradations are marked in red ($p < 0.05$). Overall, in the context of our benchmark, we find limited evidence that overthinking significantly harms performance in Qwen3 hybrid models across most answer types. However, thinking definitely reduces user utility due to increased latency. When comparing with previous-generation models (Qwen2.5-Instruct), we see clear accuracy drop for non-thinking models in numeric mode despite similar or smaller model sizes, suggesting that adding thinking capabilities to hybrid models might have compromised non-thinking mode performance.

**Analysis by Question Domains.** Figure 4a shows thinking token usage across different domains and model families, sorted by average thinking length with the highest domains at the top. The trends suggest that models generate more thinking tokens for STEM domains such as Science and Engineering, compared to domains like History. Interestingly, this occurs despite models achieving similar accuracy across these domains (Spearman $\rho = -0.46$ and $p = 0.1 > 0.05$), with little correlation between domain type and correctness. Furthermore, when examining accuracy improvements over their non-thinking counterparts, we do not find any statistically significant correlation between increased thinking and performance delta (Spearman $\rho = 0.29$ and $p = 0.33 > 0.05$). These results

Table 6: Main results on `OptimalThinkingBench` comparing open/closed thinking/non-thinking models. We also show individual results for `OverthinkingBench` and `UnderthinkingBench`, reporting accuracy, thinking tokens, and our proposed metrics. The main metrics for over, under, and optimal-thinking are $AUC_{OAA}$, accuracy, and $F_1^{otb}$ respectively. These metrics are bolded for the best performing model in each of the four categories. [†] = Hybrid models evaluated in either thinking or non-thinking mode.

| Model | OptimalThinkingBench $F_1^{otb}$ ↑ | OverthinkingBench | | | UnderthinkingBench | |
|---|---|---|---|---|---|---|
| | | Accuracy (%) ↑ | Tokens ↓ | $AUC_{OAA}$ ↑ | Accuracy (%) ↑ | Tokens ↓ |
| *Open Non-Thinking Models* | | | | | | |
| Mistral-Small-3.2-24B-2506 | 16.6 | 94.3 | 0 | 94.3 | 9.1 | 4307 |
| Llama-3.1-8B | 6.6 | 85.1 | 0 | 85.1 | 3.5 | 3811 |
| Llama-3.3-70B | 16.1 | 92.8 | 0 | 92.8 | 8.8 | 1812 |
| Llama-4-Scout | 19.1 | 95.0 | 0 | 95.0 | 10.6 | 904 |
| Llama-4-Maverick | 27.9 | 95.7 | 0 | 95.7 | 16.3 | 993 |
| Qwen2.5-7B | 9.6 | 93.6 | 0 | 93.6 | 5.1 | 1370 |
| Qwen2.5-Math-7B | 8.4 | 80.7 | 0 | 80.7 | 4.4 | 1273 |
| Qwen2.5-72B | 19.0 | 96.3 | 0 | 96.3 | 10.5 | 1174 |
| Qwen2.5-Math-72B | 15.1 | 91.8 | 0 | 91.8 | 8.2 | 1010 |
| Qwen3-1.7B[†] | 12.9 | 89.0 | 0 | 88.8 | 6.9 | 1943 |
| Qwen3-8B[†] | 24.5 | 95.9 | 0 | 95.8 | 14.0 | 2223 |
| Qwen3-14B[†] | 24.5 | 96.7 | 0 | 96.6 | 14.0 | 1585 |
| Qwen3-32B[†] | 25.8 | 96.3 | 0 | 96.2 | 14.9 | 1423 |
| Qwen3-235B-A22B[†] | 31.7 | 96.9 | 0 | 96.7 | 18.9 | 1501 |
| *Closed Non-Thinking Models* | | | | | | |
| Sonnet-4[†] | 48.3 | 97.4 | 0 | 97.4 | 32.1 | 2229 |
| GPT-4o | 17.8 | 95.3 | 0 | 95.3 | 9.8 | 694 |
| GPT-4.1 | 35.4 | 97.1 | 0 | 97.1 | 21.7 | 1846 |
| *Open Thinking Models* | | | | | | |
| Magistral-Small-2506 | 11.2 | 95.7 | 3303 | 6.4 | 42.9 | 16788 |
| R1-Distill-1.5B | 13.3 | 80.5 | 1466 | 15.2 | 11.8 | 13025 |
| DeepScaleR-1.5B-Preview | 18.8 | 82.7 | 1022 | 23.3 | 15.8 | 8617 |
| R1-Distill-7B | 24.5 | 91.5 | 1172 | 25.4 | 23.6 | 11763 |
| R1-Distill-Llama-8B | 20.7 | 93.2 | 1307 | 21.7 | 19.8 | 11113 |
| Qwen3-1.7B[†] | 24.2 | 93.8 | 1519 | 20.6 | 29.2 | 13072 |
| Qwen3-8B[†] | 24.3 | 98.1 | 1588 | 16.3 | 47.7 | 13858 |
| R1-0528-Qwen3-8B | 28.8 | 96.6 | 1926 | 24.2 | 35.7 | 15610 |
| Qwen3-14B[†] | 30.3 | 98.3 | 1373 | 21.3 | 52.4 | 12691 |
| Qwen3-32B[†] | 25.4 | 97.9 | 1415 | 16.9 | 51.0 | 12652 |
| Qwen3-235B-A22B[†] | 23.2 | 98.3 | 1632 | 14.6 | 55.5 | 12057 |
| Hunyuan-A13B | 47.1 | 96.7 | 615 | 52.2 | 42.9 | 12103 |
| GPT-OSS-20B | 57.3 | 97.1 | 467 | 72.7 | 47.3 | 8937 |
| GPT-OSS-120B | 68.3 | 97.1 | 154 | 83.3 | 57.9 | 4968 |
| *Closed Thinking Models* | | | | | | |
| Sonnet-4[†] | 64.2 | 99.3 | 706 | 71.3 | 58.3 | 14035 |
| O3 | 71.1 | 97.5 | 235 | 78.6 | 65.0 | 6273 |

Table 7: Results comparing different methods for improving optimal thinking on our benchmark. We evaluate on both `OverthinkingBench` and `UnderthinkingBench` to understand how methods developed to reduce overthinking impact underthinking and viseversa.

| Method | OptimalThinkingBench $F_1^{otb}$ ↑ | OverthinkingBench | | | UnderthinkingBench | |
|---|---|---|---|---|---|---|
| | | Accuracy (%) ↑ | Tokens ↓ | $AUC_{OAA}$ ↑ | Accuracy (%) ↑ | Tokens ↓ |
| R1-Distill-Qwen-7B | 22.8 | 85.1 | 562 | 44.9 | 15.2 | 17967 |
| + VeriThinker (Chen et al., 2025b) | 15.2 (-7.5) | 85.9 (+0.8) | 430 (-24%) | 61.5 (+16.6) | 8.7 (-6.6) | 2070 (-88%) |
| + SB-DS | 18.7 (-4.1) | 83.6 (-1.5) | 180 (-68%) | 70.5 (+25.6) | 10.8 (-4.5) | 3598 (-80%) |
| + L1 (Aggarwal & Welleck, 2025) | 24.3 (+1.5) | 84.8 (-0.3) | 562 (+0%) | 39.1 (-5.7) | 17.6 (+2.3) | 3494 (-81%) |
| + AdaptThink (Zhang et al., 2025a) | 27.3 (+4.5) | 85.4 (+0.2) | 356 (-37%) | 61.4 (+16.5) | 17.5 (+2.3) | 17176 (-4%) |
| Qwen3-8B | 34.7 | 96.3 | 854 | 30.5 | 40.3 | 19505 |
| + Model Merging (Wu et al., 2025) | 41.6 (+6.9) | 96.0 (-0.3) | 553 (-35%) | 50.9 (+20.4) | 35.1 (-5.2) | 15569 (-20%) |
| + L1 (Aggarwal & Welleck, 2025) | 34.8 (+0.1) | 95.9 (-0.4) | 560 (-34%) | 42.5 (+12.0) | 29.5 (-10.9) | 5814 (-70%) |

highlight that models cannot flexibly adjust their thinking based on the question domain, resulting in more overthinking in specific domains than in others.

**Analysis by Answer Types.** In Figure 4b, we analyze how answer types affect the amount of thinking. All models show similar behavior: they use comparable token counts for MCQ and open-ended questions while consuming substantially more tokens for numeric questions. Interestingly, unlike

Table 8: Comparison of a state-of-the-art router model (that routes between non-thinking and thinking modes based on question difficulty) with an oracle router on Qwen3 family of models to encourage optimal thinking.

| Method | OptimalThinkingBench $F_1^{otb}$ ↑ | OverthinkingBench | | | UnderthinkingBench | |
|---|---|---|---|---|---|---|
| | | Accuracy (%) ↑ | Tokens ↓ | AUC$_{OAA}$ ↑ | Accuracy (%) ↑ | Tokens ↓ |
| Qwen3-1.7B | 24.2 | 93.8 | 1521 | 20.6 | 29.2 | 26143 |
| Qwen3-1.7B-NonThink | 12.9 | 89.0 | 0 | 89.0 | 6.9 | 3886 |
| w/ Trained Router | 35.1 (+10.9%) | 91.3 | 860 | 53.6 | 26.1 | 24283 |
| Oracle Router | 43.9 | 89.0 | 0 | 89.0 | 29.2 | 26143 |
| Qwen3-8B | 24.3 | 98.1 | 1587 | 16.3 | 47.7 | 27716 |
| Qwen3-8B-NonThink | 24.5 | 95.9 | 0 | 95.9 | 14.0 | 4447 |
| w/ Trained Router | 49.1 (+24.6%) | 97.6 | 900 | 56.0 | 43.8 | 25077 |
| Oracle Router | 63.7 | 95.9 | 0 | 95.9 | 47.7 | 27716 |
| Qwen3-32B | 25.4 | 97.9 | 1423 | 16.9 | 51.0 | 25304 |
| Qwen3-32B-NonThink | 25.8 | 96.3 | 0 | 96.3 | 14.9 | 2846 |
| w/ Trained Router | 50.5 (+24.7%) | 97.2 | 815 | 55.8 | 46.1 | 22901 |
| Oracle Router | 66.6 | 96.3 | 0 | 96.3 | 51.0 | 25304 |
| Qwen3-235B-A22B | 23.2 | 98.3 | 1643 | 14.6 | 55.5 | 17133 |
| Qwen3-235B-A22B-NonThink | 31.7 | 96.9 | 0 | 96.9 | 18.9 | 3002 |
| w/ Trained Router | 53.0 (+21.3%) | 97.6 | 929 | 55.5 | 50.8 | 16691 |
| Oracle Router | 70.5 | 96.9 | 0 | 96.9 | 55.5 | 17133 |

Table 9: Results comparing different prompt variations on OptimalThinkingBench to encourage optimal thinking.

| Method | OptimalThinkingBench $F_1^{otb}$ ↑ | OverthinkingBench | | | UnderthinkingBench | |
|---|---|---|---|---|---|---|
| | | Accuracy (%) ↑ | Tokens ↓ | AUC$_{OAA}$ ↑ | Accuracy (%) ↑ | Tokens ↓ |
| | | Qwen3-1.7B | | | | |
| Standard | 24.2 | 93.8 | 1519 | 20.6 | 29.2 | 13072 |
| Step-by-Step | 19.5 (-4.7) | 93.9 (+0.1) | 1620 (+7%) | 14.6 (-6.0) | 29.4 (+0.2) | 13261 (+1%) |
| Don't Overthink | 28.8 (+4.6) | 94.2 (+0.4) | 1156 (-24%) | 30.4 (+9.8) | 27.4 (-1.8) | 12183 (-7%) |
| Only Answer | 30.8 (+6.6) | 93.9 (+0.1) | 1131 (-25%) | 32.4 (+11.8) | 29.3 (+0.1) | 12236 (-6%) |
| | | Qwen3-8B | | | | |
| Standard | 24.3 | 98.1 | 1588 | 16.3 | 47.7 | 13858 |
| Step-by-Step | 15.6 (-8.7) | 97.4 (-0.7) | 1766 (+11%) | 9.4 (-6.9) | 47.3 (-0.4) | 14400 (+4%) |
| Don't Overthink | 34.0 (+9.7) | 97.8 (-0.3) | 1233 (-22%) | 26.8 (+10.5) | 46.4 (-1.3) | 13030 (-6%) |
| Only Answer | 36.8 (+12.5) | 98.1 | 1247 (-21%) | 29.7 (+13.4) | 48.4 (+0.7) | 13149 (-5%) |
| | | Qwen3-14B | | | | |
| Standard | 30.3 | 98.3 | 1373 | 21.3 | 52.4 | 12691 |
| Step-by-Step | 19.8 (-10.5) | 97.9 (-0.4) | 1529 (+11%) | 12.1 (-9.2) | 53.0 (+0.6) | 13078 (+3%) |
| Don't Overthink | 39.2 (+8.9) | 98.1 (-0.2) | 1051 (-23%) | 31.4 (+10.1) | 52.1 (-0.3) | 12058 (-5%) |
| Only Answer | 42.3 (+12.0) | 97.9 (-0.4) | 955 (-30%) | 35.7 (+14.4) | 51.8 (-0.6) | 11817 (-7%) |

OvT-Math, numeric questions in OvT-General are primarily fact based (See Appendix subsection D.4 for examples). A potential reason for this difference could be due to the increased computational complexity demanded by the numeric questions. However, as shown in Appendix subsection D.2, we evaluate the accuracy of 5 different models and find no statistically significant difference in accuracy compared to non-thinking models for the numeric domain in 4 out of 5 cases. This finding suggests that mathematical tokens in prompts trigger more extensive thinking (possibly because of the heavy reliance on mathematical tasks in post-training), regardless of the underlying complexity of the questions.

**Analysis by Number of Distractors in MCQs.** Finally, in Figure 10, we analyze how overthinking varies with the number of options for multiple choice questions. The figure shows the average number of thinking tokens versus the number of multiple-choice questions averaged across all 5 Qwen3 models. In particular, we augment the original multiple-choice questions in OverthinkingBench by adding completely irrelevant options in the questions. Interestingly, despite being completely irrelevant, we see a clear rise in thinking tokens with an increasing number of options. In particular, we see an almost linear ($R^2 = 0.94$) increase of 42 tokens per option, indicating how irrelevant distractors can lead to overthinking in models.

Table 10: Delta accuracy between thinking and non-thinking mode for different models and answer types. Values show delta accuracy (%). Dark green indicates statistically significant positive changes, red indicates statistically significant negative changes ($p < 0.05$).

| Model | MCQ | Numeric | Open-ended | Open-ended-long | Average |
|---|---|---|---|---|---|
| Qwen3-1.7B | 2.2% | 4.5% | 4.7% | 3.5% | 3.7% |
| Qwen3-8B | 0.8% | 2.3% | 1.0% | -0.5% | 0.9% |
| Qwen3-14B | 0.0% | 0.9% | 0.9% | -0.3% | 0.4% |
| Qwen3-32B | -0.2% | 1.0% | -0.9% | -0.3% | -0.1% |
| Qwen3-235B-A22B | -1.1% | 1.6% | 0.1% | 0.3% | 0.2% |
| *vs Qwen2.5-7B* | | | | | |
| Qwen3-8B-nonthink | 0.8% | -0.0% | 2.6% | 1.4% | 1.2% |
| *vs Qwen2.5-72B* | | | | | |
| Qwen3-32B-nonthink | 0.2% | 2.8% | 1.0% | -0.5% | -0.5% |
| Qwen3-235B-A22B-nonthink | 0.5% | 2.1% | 1.0% | -0.6% | -0.3% |

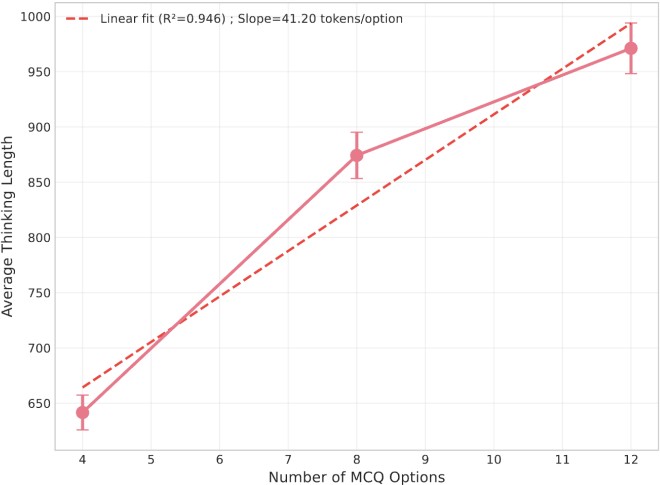

Figure 10: Results showing how amount of overthinking varies with the number of options for multiple choice questions. Despite most options being distractors, there is almost a linear increase in overthinking with an increasing number of options.

### D.3 HOW SCALING MODEL SIZE AFFECTS OVERTHINKING?

We examine how overthinking varies with model sizes across the Qwen3 family of models on `OvT-General`. The results reveal that while there is no clear trend, thinking token usage generally increases with model size, despite accuracy remaining relatively constant across models from 8B onwards. Specifically, thinking tokens increase from 750 tokens (1.7B) to 950 tokens (235B), while accuracy fluctuates between 86.1% and 86.8% for models 8B and larger. The increased thinking does not translate to improved performance, as all models starting from 8B parameters achieve similar accuracies. This suggests that larger models may be exhibiting unnecessary verbosity in their reasoning without corresponding performance benefits, highlighting an area for optimization in model training or inference procedures.

### D.4 QUALITATIVE ANALYSIS OF OVERTHINKING AND UNDERTHINKING

In this section, we qualitatively compare how overthinking could hurt performance on `OverthinkingBench` and how non-thinking models can underthink and rely on heuristics in `UnderthinkingBench`. However, to compare two models of similar accuracy, naively selecting

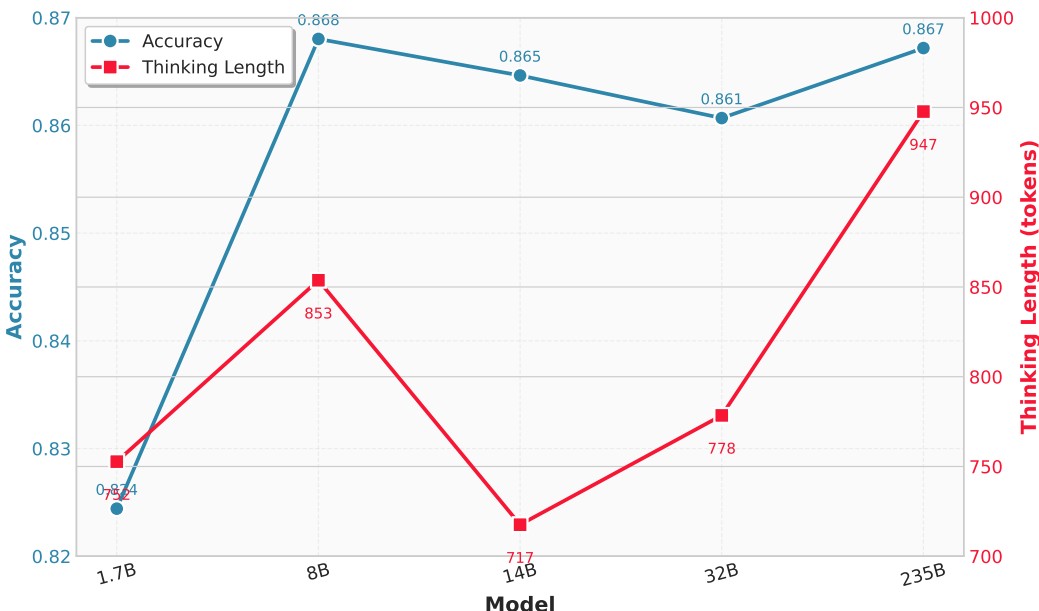

Figure 11: Results for how overthinking and accuracy vary with changing model size for Qwen3 family of models.

questions where model A does better than model B is not appropriate due to the stochastic nature of models. For a fair comparison, we generate 128 responses for each model and only consider situations where the performance difference is statistically significant. In Examples 1, 2, and 3, we show three instances where overthinking by models leads to an incorrect answer. In particular, we notice a common phenomenon where the model initially comes up with the correct answer but then overthinks either because of conflicting information or incorrect reasoning. In Examples 4 and 5, we show cases where non-thinking models underthink. In one such case the model says that it would use BFS; however, it declared its first attempt as the correct one, without any self-verification or exploration of other solutions.

**Example 1: Overthinking**

**Question:** What is the name of the layer in the atmosphere that protects the Earth from UV radiation?

**Model Thinking Process:**

Alright, the question is asking about the layer in the atmosphere that protects the Earth from UV radiation. I know that the Earth's atmosphere has several layers, and one of them is responsible for absorbing and blocking harmful UV rays.

First, I recall the layers of the atmosphere from the surface outward: 1. Troposphere 2. Stratosphere 3. Mesosphere 4. Thermosphere 5. Exosphere

Now, which one of these is responsible for protecting Earth from UV radiation? I remember that **the ozone layer is important for this**, but the ozone layer is part of a specific atmospheric layer.

I think the ozone layer is located in the stratosphere. But I'm not entirely sure, so I recall that the stratosphere is the second layer from the surface, and it's where the ozone layer resides. **The ozone layer absorbs most of the Sun's harmful UV radiation, protecting life on Earth.**

**But just to be sure, I recall that the ozone layer is in the lower part of the stratosphere. So, the stratosphere is the layer that contains the ozone layer, which protects Earth from UV radiation.**

**But wait, is the stratosphere itself the protector, or is it specifically the ozone layer within the stratosphere? The question asks for the name of the layer in the atmosphere, not the specific component (ozone). So, the layer is the stratosphere.**

*[. . . truncated . . . ]*

**But, I recall that sometimes people refer to the ozone layer itself as the protective layer. However, the question specifies "the layer in the atmosphere," and the ozone layer is a part of the stratosphere. But in common terminology, people might say the ozone layer is the protective layer. But the ozone layer isn't one of the main atmospheric layers (troposphere, stratosphere, etc.), it's a region within the stratosphere.**

*[. . . truncated . . . ]*

**But, I think I might be confusing the terminology. The ozone layer is part of the stratosphere, but the stratosphere as a whole is not solely responsible for protecting from UV radiation—it's the ozone within it. However, the question asks for the layer in the atmosphere, not the specific component. So, the stratosphere is the layer that contains the ozone layer, which protects Earth from UV radiation.**

*[. . . truncated . . . ]*

After all this thinking, I conclude that the correct answer is the stratosphere, because it is the atmospheric layer that contains the ozone layer, which protects Earth from UV radiation.

**Final Response:**

The layer in the atmosphere that protects the Earth from UV radiation is the stratosphere, as it contains the ozone layer which absorbs and scatters the harmful UV rays.

**Answer:** $\boxed{Stratosphere}$

**Evaluation:** × **Incorrect**

**Expected Answer:** Ozone Layer

**Analysis:** The model demonstrates clear **overthinking behavior** by repeatedly second-guessing itself and going in circles about whether the answer should be "stratosphere" or "ozone layer." **The model initially had the correct knowledge** that the ozone layer protects from UV radiation, but the excessive deliberation led to an incorrect final answer due to confusion over terminology distinctions.

**Example 2: Overthinking with Wrong Conclusion**

**Question:** Which pop star is known for her hit song 'Thank U, Next'?

**Model Thinking Process:**

Okay, so I need to figure out which pop star is known for the hit song "Thank U, Next." Let me start by recalling what I know about this song. I think it's a recent hit, maybe from the past few years. The title "Thank U, Next" sounds familiar, maybe from a music video or some awards show.

First, I should remember if there's a female artist associated with this song. The name "Thank U, Next" might be a bit of a play on words, maybe referencing a previous song or artist. Wait, there's a song called "Thank You, Next" by a pop artist. Let me think... I recall that Lady Gaga had a song called "Thank You, Next," but I'm not sure if that's the same. Wait, no, maybe it's another artist.

*[. . . truncated . . . ]*

**Wait, I remember that Ariana Grande had a song called "Thank U, Next."** Let me check that. **No, actually, I think the song is by a different artist.** Oh! Maybe it's by a singer named Megan Thee Stallion? No, I think that's more about her "Don't Stop" or something else.

**Wait, I'm getting confused. Let me try to recall. There's a song that was a big hit in 2018, and the artist is a female pop star.** *The song "Thank U, Next" was written by a singer who also had hits like "Hello" and "All I Want." Wait, that sounds like Adele, but she didn't have a song with that title.*

*Hold on, maybe it's by the singer who had a song called "Love Me Like You" and "Back to Black." No, that's Amy Winehouse.*

**Wait, I think the answer is Lady Gaga. She had a song called "Thank You, Next." But I'm not sure. Alternatively, maybe it's by a newer artist.**

*Wait, I think I remember that the song "Thank U, Next" was released by a pop star named Ariana Grande. Wait, no, Ariana's song is called "Thank U, Next." Let me check my memory.*

**Wait, no, I think the correct answer is that the song "Thank U, Next" is by the pop star Lady Gaga. But I'm not 100% sure. Alternatively, maybe it's by someone else.**

*[. . . truncated . . . ]*

**Wait, another approach: The song "Thank U, Next" was a number one hit in the US. The artist is a pop star who had previous hits.** *Let me think of the artist who had a song called "Poker Face" and "Bad Romance." That's Lady Gaga. So if she had a song called "Thank U, Next," that would make sense.*

*Wait, but I've heard that the song "Thank U, Next" is actually by a different artist. Maybe it's by someone like Ariana Grande?*

**I think I need to make a decision here. Based on what I remember, the song "Thank U, Next" is by Lady Gaga. So the answer is Lady Gaga.**

**Final Response:**

The song "Thank U, Next" is a well-known hit that was released in 2018. It is associated with the pop star Lady Gaga, who is recognized for her powerful vocals, iconic music videos, and a string of successful hits such as Poker Face, Bad Romance, and Just Dance. The song was a significant success, reaching number one on the Billboard Hot 100 and receiving widespread acclaim for its emotional depth and lyrical honesty.

**Answer:** | Lady Gaga |

**Evaluation:** × **Incorrect**

**Expected Answer:** Ariana Grande

**Analysis:** The model shows severe **overthinking behavior** and *confusion between artists*. **It correctly identified Ariana Grande initially** but then *incorrectly attributed Lady Gaga's other hits to this song*. The model's excessive deliberation and self-doubt led it away from the correct answer despite having the right information early in its reasoning process.

**Example 3: Overthinking with Conflicting Information**

**Question:** What is the number of time zones in Russia?

**Model Thinking Process:**

Okay, so I need to figure out how many time zones Russia has. Let me start by recalling what I know. **Russia is a huge country, spanning a lot of east-west distance. I remember that time zones are generally based on longitudinal divisions, with each time zone roughly covering 15 degrees of longitude, which corresponds to one hour.**

First, I think Russia is in the northern hemisphere, and it's located between Europe and Asia. I know that the country is so large that it crosses multiple time zones. But how many exactly?

**Wait, I think I've heard that Russia has 11 time zones. But I'm not entirely sure.** Let me try to remember or think through the details. **The main time zone in Russia is Moscow Time, which is UTC+3. But then there's also other zones. For example, there's Yekaterinburg Time (UTC+5), Omsk Time (UTC+6), Krasnoyarsk Time (UTC+7), Irkutsk Time (UTC+8), Yakutsk Time (UTC+9), Vladivostok Time (UTC+10), and Magadan Time (UTC+11).** That's like 7 time zones. **But wait, I thought there were 11. Maybe there are more?**

*[. . . truncated . . . ]*

**That's 9. But I thought there were 11. Wait, maybe there are more splits. Let me check.** Oh, I think there's also the time zone in the Far East, but maybe some of these zones are split further? Or maybe I'm mixing up with other countries.

**Alternatively, perhaps the answer is 11, but I need to confirm. Let me think about the history. Russia used to have more time zones, but some were merged. For example, in 2014, they reduced the number of time zones from 11 to 9.** Wait, but in 2014, they actually changed the time zones, reducing the number. So maybe the current number is 9. But I need to be careful here.

*[. . . truncated . . . ]*

*But some sources say 11. Maybe the confusion comes from the fact that before 2014, there were more. Let me check my memory. In 2014, Russia reduced the number of time zones from 11 to 9 by merging some. So the current number is 9. However, there's a possibility that some sources still refer to the old number. But according to recent information, it's 9. But I need to be sure.*

**Example 3: Overthinking with Conflicting Information (continued)**

**Model Thinking Process:**

> **Wait, another way to think: the country spans from the western part (UTC+3) to the eastern part (UTC+12), which is a difference of 9 hours. Since each time zone is 1 hour, that would be 9 time zones. But wait, the difference between UTC+3 and UTC+12 is 9 hours, so that would mean 9 time zones. But earlier I thought there were 11.**
>
> *[. . . truncated . . . ]*
>
> *Wait, this is getting too confusing. Alternatively, perhaps the answer is that Russia currently has 11 time zones, but some sources might say 9 or 10. However, according to recent information, after the 2014 adjustments, Russia has 9 time zones.*
>
> **I think the commonly accepted answer is that Russia has 11 time zones, but due to the 2014 changes, some sources say 9 or 10. However, the correct answer is 11, as the country spans from UTC+2 to UTC+12, which is 10 time zones, but including Kaliningrad as UTC+2, it's 11.**
>
> **Therefore, despite the 2014 changes, the number of time zones in Russia is 11. However, some sources might say 9 or 10 due to the adjustments. But the correct answer is 11.**

**Final Response:**

> Russia spans a vast territory across multiple longitudinal regions, leading to a significant number of time zones. Historically, the country had 11 time zones, which stretched from UTC+2 (Kaliningrad, an exclave in the west) to UTC+12 (Kamchatka, in the far east).
>
> However, in 2014, Russia implemented a major reform to reduce the number of time zones from 11 to 9. This involved merging some of the previously distinct time zones.
>
> Despite this reduction, the total number of time zones currently recognized in Russia is 9.
>
> **Answer:** $\boxed{9}$

**Evaluation:** × **Incorrect**

**Expected Answer:** 11

**Analysis:** The model demonstrates **extensive deliberation** and *conflicting information processing*. While it **correctly mentions the 2014 time zone reforms**, it gets confused between historical and current numbers. The model shows classic overthinking by going back and forth between 9 and 11, ultimately providing an inconsistent conclusion (claiming 11 is correct but answering 9). The correct answer is 11 time zones as of current Russian law.

**Example 4: Underthinking in Pathfinding**

**Question:** Find the shortest path from * to the destination in the grid (output as sequence of directions):

```
X X O O O O *
X O X X O X O
X H X O O O O
X O O X O O X
O O O O O O X
```

where H represents the destination point

**Model Thinking Process:**

We are given a grid and need to find the shortest path from the start point (*) to the destination point (H), moving only through open cells (O), and not passing through blocked cells (X). Movement is allowed in four directions: up, down, left, right.

**Let me locate the start (*) and destination (H) points in the grid.** - Start (*) is at position (0, 6) - Destination (H) is at position (2, 1)

**I need to use BFS to find the shortest path. Let me just trace a path quickly.**

**BFS proceeds as follows: - From (0,6) → (1,6) (down) - From (1,6) → (2,6) (down) - From (2,6) → (2,5) (left) - From (2,5) → (2,4) (left) - From (2,4) → (2,3) (left) - From (2,3) → (2,2) (left) - From (2,2) → (2,1) (left)**

**So the direction sequence is: down down left left left left left**

**Final Response:**

We are given a grid and need to find the shortest path from the start point (*) to the destination point (H), moving only through open cells (O).

Step 1: Find Start and Destination - Start (*) is at position (0, 6) - Destination (H) is at position (2, 1)

Step 2: BFS to Find the Shortest Path BFS proceeds as follows: - From (0,6) → (1,6) (down) - From (1,6) → (2,6) (down) - From (2,6) → (2,5) (left) - From (2,5) → (2,4) (left) - From (2,4) → (2,3) (left) - From (2,3) → (2,2) (left) - From (2,2) → (2,1) (left)

**Answer:** | down down left left left left left |

**Evaluation:** × **Incorrect**

**Expected Answer:** down down left left down down left left left up up

**Analysis:** The model shows classic **underthinking behavior** by claiming to use BFS but never actually implementing or verifying it. It **assumes a path exists without checking cell validity** and **follows the first viable route without exploring alternatives**. The model **skips systematic exploration** and **doesn't verify the optimality** of its solution.

**Example 5: Underthinking in Arithmetic**

**Question:** Solve: ((0xe293 * 0xcd5d) + (0xd7b7 + 0x2442))

**Model Thinking Process:**

I need to solve ((0xe293 * 0xcd5d) + (0xd7b7 + 0x2442)). **I'll just convert to decimal, do the arithmetic, and convert back.**

Step 1: Compute 0xe293 * 0xcd5d **Convert both values to decimal: - 0xe293 in hex = 57987 in decimal - 0xcd5d in hex = 52573 in decimal**

**Now compute the product: 57987 × 52573 = 3050347251**

**Convert back to hexadecimal: 3050347251 in hex = 0x74b1f51f**

Step 2: Compute 0xd7b7 + 0x2442 **Convert both values to decimal: - 0xd7b7 = 54695 - 0x2442 = 9282**

**Add them: 54695 + 9282 = 63977**

**Convert back to hexadecimal: 63977 in hex = 0xf9d1**

Step 3: Add the two results **Now add: 0x74b1f51f + 0xf9d1 = 0x74b2eef0**

**Final Response:**

To solve the expression: ((0xe293 * 0xcd5d) + (0xd7b7 + 0x2442))

We will compute each part step by step using hexadecimal arithmetic.

Step 1: Compute 0xe293 * 0xcd5d Convert both values to decimal: - 0xe293 in hex = 57987 in decimal - 0xcd5d in hex = 52573 in decimal

Now compute the product: 57987 × 52573 = 3050347251 Convert back to hexadecimal: 3050347251 in hex = 0x74b1f51f

Step 2: Compute 0xd7b7 + 0x2442 Convert both values to decimal: - 0xd7b7 = 54695 - 0x2442 = 9282

Add them: 54695 + 9282 = 63977 Convert back to hexadecimal: 63977 in hex = 0xf9d1

Step 3: Add the two results Now add: 0x74b1f51f + 0xf9d1 = 0x74b2eef0

**Answer:** $\boxed{0x74b2eef0}$

**Evaluation:** ✗ **Incorrect**

**Expected Answer:** 0xb5c30260

**Analysis:** The model demonstrates **insufficient verification** of its calculations. It **doesn't double-check the decimal conversions**, leading to a cascade of errors. The initial conversions for '0xe293' (which is '58003') and '0xd7b7' (which is '55223') are incorrect. This reliance on unverified intermediate steps leads to an incorrect final result. A careful recalculation would have shown the correct answer should be '0xb5c30260'.

## D.5 AUTOMATED ANALYSIS OF UNDERTHINKING PATTERNS

To further quantify underthinking behaviors, we conduct an automated analysis of reasoning chains on `UnderthinkingBench` using a two-step prompting approach with o3. Given an incorrect chain of thought from a model, we first prompt o3 to segment the reasoning into coherent, non-overlapping steps based on subgoal switches, method changes, or conclusion points. In the second step, we provide o3 with both the segmented incorrect chain and a correct chain of thought as reference, prompting it to classify each segment for underthinking patterns. We define underthinking as instances where models fail to carry out necessary reasoning, verification, or exploration—distinct from simple numerical or symbolic errors.

We categorize underthinking into several subtypes: (1) *no verification*—failing to verify intermediate results or the final answer; (2) *premature conclusion*—stopping reasoning early or accepting assumptions without justification; (3) *missing case analysis*—skipping required case analysis or not exploring all relevant possibilities; (4) *promised method not executed*—claiming to use a method or algorithm but not actually executing it; and (5) *ignored contradiction*—ignoring clear contradictions introduced by earlier steps. Each identified underthinking segment is assigned a criticalness score (0–9) indicating its contribution to the final incorrect answer.

**Quantitative Results.** Table 11 presents the results of our automated analysis across both non-thinking and thinking models. We observe several key patterns. First, non-thinking models exhibit underthinking more frequently, with 87.2% of samples containing at least one underthinking segment on average, compared to 70.5% for thinking models. Second, while thinking models have a lower rate of samples with underthinking, they are not immune to this problem—even large thinking models like Qwen3-235B show underthinking in 62.4% of cases. Third, distilled reasoning models (R1-Distill variants) show surprisingly high underthinking rates (84–91%), suggesting that distillation may not fully transfer robust reasoning verification behaviors. Note that the average number of underthinking steps is higher for thinking models because their longer reasoning chains lead to more segments; thus, Fraction UT provides a more reliable comparison across model types.

**Error Type Analysis.** Table 12 shows the distribution of underthinking subtypes across model categories. The most striking difference is in verification failures: non-thinking models fail to verify their reasoning in 44.6% of underthinking cases, compared to 34.6% for thinking models. Conversely, thinking models more frequently exhibit premature conclusions (21.9% vs 17.6%), suggesting they sometimes abandon complex reasoning prematurely rather than failing to initiate verification. Both model types show similar rates of missing case analysis (∼14%) and promised-but-not-executed methods (∼10–11%). These patterns suggest that non-thinking models primarily underthink by skipping verification steps, while thinking models are more prone to giving up on complex reasoning paths.

**Example Analysis Outputs.** Below we show two example outputs from our automated analysis, illustrating how underthinking manifests in model reasoning:

```
Example 1: No Verification (GPT-4o)

Segment ID: 2
Segment Snippet:  ``Step 1:  Look for neighboring tokens that can
be rewritten...  2.  **B# #B** → no rewrite (no '#' facing each
other)...''
Underthinking:  True
Error Subtype:  no_verification
Criticalness:  8
Error Snippet:  ``B# #B → no rewrite (no '#' facing each other)''
Reasoning:  The model claims that the pair ``B# #B'' has no facing
hashes, but by its own stated rules a right-hash (B#) followed by
a left-hash (#B) must be rewritten to nothing.  Failing to notice
this shows it did not verify the facing-hash condition it described,
leading to an incorrect state that corrupts all further steps.
```

These examples illustrate the two dominant underthinking patterns: non-thinking models often skip verification of their stated rules (Example 1), while thinking models sometimes prematurely abandon

Table 11: Automated analysis of underthinking patterns in model reasoning chains on `UnderthinkingBench`. We use a two-step prompting approach with o3 to segment incorrect chains of thought and classify each segment for underthinking. "At Least One UT" indicates the percentage of samples with at least one underthinking segment. "Avg. UT Steps" shows the average number of underthinking segments per sample. "Fraction UT" is the average fraction of segments classified as underthinking.

| Model | At Least One UT (%) ↓ | Avg. UT Steps ↓ | Fraction UT (%) ↓ |
|---|---|---|---|
| *Non-Thinking Models* | | | |
| GPT-4o | 87.6 | 1.67 | 46.4 |
| GPT-4.1 | 89.5 | 1.60 | 69.3 |
| Sonnet-4 (non-think) | 67.0 | 1.72 | 32.5 |
| Mistral-Small-3.2-24B | 90.5 | 2.99 | 51.0 |
| Llama-3.1-8B-Instruct | 96.1 | 2.25 | 64.1 |
| Llama-3.3-70B-Instruct | 93.5 | 3.02 | 55.8 |
| Llama-4-Scout-17B | 87.9 | 3.52 | 47.8 |
| Llama-4-Maverick-17B | 80.9 | 2.87 | 40.9 |
| Qwen2.5-7B-Instruct | 94.4 | 2.01 | 70.1 |
| Qwen2.5-Math-7B-Instruct | 87.5 | 2.27 | 57.5 |
| Qwen2.5-72B-Instruct | 87.4 | 2.00 | 58.4 |
| Qwen2.5-Math-72B-Instruct | 83.7 | 2.27 | 46.3 |
| Qwen3-1.7B (non-think) | 96.8 | 2.18 | 49.3 |
| Qwen3-8B (non-think) | 91.5 | 2.06 | 47.0 |
| Qwen3-14B (non-think) | 88.7 | 1.78 | 46.8 |
| Qwen3-32B (non-think) | 84.8 | 1.88 | 42.5 |
| Qwen3-235B-A22B (non-think) | 74.3 | 1.81 | 32.9 |
| *Average (Non-Thinking)* | *87.2* | *2.23* | *50.5* |
| *Thinking Models* | | | |
| Sonnet-4 | 53.2 | 1.21 | 24.3 |
| Magistral-Small-2506 | 68.0 | 2.37 | 36.1 |
| R1-Distill-1.5B | 91.2 | 3.68 | 64.7 |
| R1-Distill-7B | 84.0 | 3.25 | 53.1 |
| R1-Distill-Llama-8B | 90.1 | 3.33 | 56.6 |
| DeepScaleR-1.5B | 85.9 | 3.24 | 58.9 |
| R1-0528-Qwen3-8B | 63.6 | 2.09 | 31.3 |
| Qwen3-1.7B | 72.2 | 2.58 | 43.0 |
| Qwen3-8B | 63.4 | 2.14 | 33.4 |
| Qwen3-14B | 58.6 | 1.95 | 30.6 |
| Qwen3-32B | 57.9 | 2.02 | 30.8 |
| Qwen3-235B-A22B | 62.4 | 2.23 | 32.0 |
| Hunyuan-A13B | 65.9 | 2.41 | 36.9 |
| *Average (Thinking)* | *70.5* | *2.50* | *40.9* |

complex computations (Example 2). Both patterns contribute to incorrect final answers and highlight areas for improvement in future model development.

Table 12: Distribution of underthinking error types (%) across model categories. "Verification" indicates failure to verify intermediate results. "Premature" refers to prematurely stopping reasoning or accepting assumptions. "Missing" denotes skipped case analysis. "Promised" means claiming to use a method but not executing it. "Ignore" refers to ignoring contradictions.

| Model | Verif. | Premat. | Missing | Promised | Ignore | Other |
|---|---|---|---|---|---|---|
| *Non-Thinking Models* | | | | | | |
| GPT-4o | 54.6 | 17.5 | 10.6 | 9.5 | 1.8 | 6.0 |
| Llama-3.3-70B-Instruct | 40.3 | 16.8 | 19.0 | 13.9 | 3.6 | 6.4 |
| Qwen3-235B (non-think) | 48.8 | 18.8 | 12.1 | 12.7 | 3.4 | 4.2 |
| *Avg. Non-Thinking* | *44.6* | *17.6* | *14.2* | *10.3* | *4.5* | *8.8* |
| *Thinking Models* | | | | | | |
| Qwen3-8B | 35.4 | 24.5 | 14.4 | 13.0 | 6.4 | 6.3 |
| Qwen3-235B | 33.5 | 27.2 | 13.4 | 15.0 | 5.0 | 6.0 |
| Sonnet-4 | 37.7 | 22.7 | 15.8 | 15.2 | 2.5 | 5.9 |
| R1-Distill-7B | 36.7 | 19.7 | 12.8 | 9.7 | 8.5 | 12.6 |
| *Avg. Thinking* | *34.6* | *21.9* | *14.3* | *11.2* | *7.5* | *10.5* |

---

**Example 2: Premature Abandonment (Qwen3-235B)**

**Segment ID:** 8

**Segment Snippet:** ``Let's process the leftmost first: 1-2... This is getting very time-consuming...''

**Underthinking:** True

**Error Subtype:** premature_abandonment

**Criticalness:** 7

**Error Snippet:** ``This is getting very time-consuming.''

**Reasoning:** The model starts another manual reduction, quickly becomes confused, and explicitly gives up because it is ``getting very time-consuming'' instead of completing or validating the computation. Abandoning the necessary step leads directly to an incorrect conclusion.

