# OpenReview forum: "OptimalThinkingBench: Evaluating Over and Underthinking in LLMs"
_ICLR.cc/2026/Conference — ICLR 2026 Poster_

### Official Review · Reviewer_gas8 · 2025-10-22

**Soundness:** 2
**Presentation:** 2
**Contribution:** 2
**Rating:** 2
**Confidence:** 4

**Summary:**

This paper introduces a unified benchmark that jointly evaluates overthinking and underthinking in LLMs and also encourages the development of optimally-thinking models that balance performance and efficiency. This benchmark comprises two sub-benchmarks: OverthinkingBench, featuring simple math and general queries in 72 domains, and UnderthinkingBench, containing 11 challenging reasoning tasks along with harder math problems. The author evaluate 33 different thinking and non-thinking models and show that no model is able to optimally think on our benchmark. Thinking models often overthink for hundreds of tokens on the simplest user queries without improving performance. In contrast, large nonthinking models underthink, often falling short of much smaller thinking models.

**Strengths:**

This paper introduces a unified benchmark that jointly evaluates overthinking and underthinking in LLMs and also encourages the development of optimally-thinking models that balance performance and efficiency. Different thinking and non-thinking models and show that no model is able to optimally think on our benchmark.

**Weaknesses:**

1. The paper's central claim is that "no model is able to optimally think." However, this conclusion is drawn from a pool of models that may not represent the current state-of-the-art in reasoning and instruction-following. Top-tier commercial models like OpenAI's GPT-4/GPT-4o, Anthropic's Claude 3 family, and Google's Gemini family are specifically engineered to handle a wide distribution of task difficulties. More commercial models should be considered.
2. It seams that many other simple benchmarks can work as the no-thinking parts(many instance like MATH500), and many hard benchmarks can work as the thinking parts(like HLE math parts). The authors should highlight more unique features of this benchmark, as it currently appears highly replaceable.
3. Limited exploration on "optimal thinking". The explored methods are not detailed in the summary, but they might be simple approaches like static routing or basic prompting. More advanced techniques exist, such as dynamic router models that predict the required reasoning depth, or using the LLM itself to decide if it needs to "think" (i.e., use a COT prompt). Without comparing against these more sophisticated strategies, the conclusion that the trade-off is unavoidable seems premature.

**Questions:**

Just as the statements on weakness.

---

> ### Author Response · Authors · 2025-11-20
>
> > The paper's central claim is that "no model is able to optimally think." However, this conclusion is drawn from a pool of models that may not represent the current state-of-the-art in reasoning and instruction-following. Top-tier commercial models like OpenAI's GPT-4/GPT-4o, Anthropic's Claude 3 family, and Google's Gemini family are specifically engineered to handle a wide distribution of task difficulties. More commercial models should be considered.
>
> We would like to clarify some misunderstandings here. As shown in Table 1, we have, in fact, evaluated frontier commercial models such as **GPT-4o, GPT-4.1, o3, and Claude Sonnet-4** in both thinking and non-thinking modes, alongside many other state-of-the-art open-source models of varying sizes and families. Therefore, the statement that our pool of models "may not represent the current state-of-the-art in reasoning and instruction-following" is factually incorrect.
>
> > It seems that many other simple benchmarks can work as the no-thinking parts(many instance like MATH500), and many hard benchmarks can work as the thinking parts(like HLE math parts). The authors should highlight more unique features of this benchmark, as it currently appears highly replaceable.
> - **Unified Evaluation:** As we discuss in the Introduction and Section 4, it is not possible to trivially combine benchmarks because prior works use disparate evaluation setups and unique metrics. `OptimalThinkingBench' solves this by providing a standardized $F_1^{\text{otb}}$ metric to jointly track progress.
> - **Controlled Construction:** `OverthinkingBench' requires rigorous filtering (Figure 5) to ensure questions are unambiguous and simple enough that thinking is truly unnecessary, which standard datasets do not guarantee. Further, unlike other benchmarks such as HLE that often rely on knowledge-intensive tasks, the UT-Reasoning split ensures that we can measure purely reasoning capability due to its algorithmic nature.
> - **Existing Benchmarks:** Nonetheless, we do include Level 1-2 problems from the MATH dataset as the non-thinking subset and AIME’25/HMMT’25 as the thinking subset of our benchmark (Section 2.1, 2.2), while expanding coverage to 72 diverse domains beyond just math.
>
> > Limited exploration on "optimal thinking". The explored methods are not detailed in the summary, but they might be simple approaches like static routing or basic prompting. More advanced techniques exist, such as dynamic router models that predict the required reasoning depth, or using the LLM itself to decide if it needs to "think" (i.e., use a COT prompt). Without comparing against these more sophisticated strategies, the conclusion that the trade-off is unavoidable seems premature.
>
> Let us clarify a misunderstanding here. We have extensively evaluated methods that you have mentioned, such as dynamic routing based on problem difficulty (Table 3), models trained with RL to explicitly decide when to think (AdaptThink in Table 2), various state-of-the-art methods shown to reduce overthinking (Table 2), and various prompting strategies (Table 4). The results consistently point to the fact that these current methods are not effective at balancing the trade-off (Section 3.3). If the reviewer has concrete suggestions for any additional method to be evaluated, we are happy to do so.
>
> ---
>
> Thank you for your review. We hope our responses have addressed your concerns. If you think there are still some concerns, we look forward to addressing them as well! Alternatively, if you feel all your concerns are addressed, we would appreciate it if you could reflect that in your evaluation!

---

> > ### Comment · Reviewer_gas8 · 2025-11-23
> > **Reply the rebuttal**
> >
> > I appreciate that the authors’ restatement has addressed the misunderstanding, and I am willing to raise my score from 2 to 4. However, I would like to see further discussion on whether this benchmark reveals the existence and extent of underthinking (as opposed to the relatively simpler overthinking) among current mainstream models, including both open-source and closed-source ones

---

> > > ### Author Response · Authors · 2025-12-02
> > >
> > > We are glad that our clarifications have addressed the misunderstanding, and we thank the reviewer for increasing their score.
> > >
> > > > whether this benchmark reveals the existence and extent of underthinking among current mainstream models, including both open-source and closed-source ones
> > >
> > > Yes, our work *explicitly reveals both the existence and extent of underthinking*. We would like to clarify that
> > > 1.) Our work has extensively tested underthinking across 33 different open-source, closed-source, thinking, and non-thinking models, and has found consistent underthinking even for frontier models, which forms part of our core results (Section 3.2, Table 1, and Table 6 in Appendix D). We have further studied underthinking across domains (Figure 3).
> > > 2.) We have extensively evaluated methods to address underthinking, including dynamic routing based on problem difficulty (Table 3), models trained with RL to explicitly decide when to think, such as AdaptThink (Table 2), various state-of-the-art efficient reasoning methods (Table 2), and various prompting strategies (Table 4). The results consistently show that these current methods are not effective at balancing the accuracy-efficiency trade-off (Section 3.3), often because of underthinking.
> > > 3.) We also conducted qualitative analysis (Section 3.4 and Appendix D.4), showcasing how "models rely on heuristics without verification—claiming to use algorithms like BFS while actually taking the first plausible path without systematic exploration or validation" (Examples 4 and 5).
> > >
> > > We therefore want to clarify that analyzing the extent of underthinking already forms the bedrock of our contributions.
> > >
> > > > more analysis of **extent** of underthinking
> > >
> > > *Nonetheless, if the reviewer still thinks any analysis is missing, we have added a new quantitative analysis in Appendix D.5*; we provide a brief summary below.
> > > The additional quantitative analysis identifies key behaviors of underthinking in reasoning chains across all models. We used an automated two-step prompting method using o3 (the best-performing model), with access to privileged information (both correct and incorrect chains of thought), to assess segments of reasoning chains and identify whether underthinking occurs. Each segment is labelled into one of 5 types of underthinking: We define five main types: (1) no verification, (2) premature conclusion, (3) missing case analysis, (4) promised method not executed, (5) ignored contradiction, and others. Specifically, our analysis reveals that non-thinking models underthink frequently, for instance, GPT-4o underthinks at least once in 87\% of cases. Thinking models also underthink substantially, with Qwen3-235B underthinking at least once in 62\% of cases. We further find that non-thinking models fail to verify their incorrect steps roughly 50\% of the time, while often missing necessary case analysis. On the other hand, thinking models often prematurely abandon or conclude their incorrect reasoning steps (e.g., 25\% of the time).
> > >
> > > This analysis further highlights the **existence**, **extent**, and **nature** of underthinking on top of the already comprehensive experiments in our paper.
> > >
> > > A short snippet of the full table is provided here:
> > >
> > > | Model | Verif. | Premat. | Missing | Promised | Ignore | Other |
> > > |---|---|---|---|---|---|---|
> > > | *Non-Thinking Models* | | | | | | |
> > > | GPT-4o | 54.6 | 17.5 | 10.6 | 9.5 | 1.8 | 6.0 |
> > > | Llama-3.3-70B-Instruct | 40.3 | 16.8 | 19.0 | 13.9 | 3.6 | 6.4 |
> > > | Qwen3-235B (non-think) | 48.8 | 18.8 | 12.1 | 12.7 | 3.4 | 4.2 |
> > > | *Avg. Non-Thinking* | *44.6* | *17.6* | *14.2* | *10.3* | *4.5* | *8.8* |
> > > | *Thinking Models* | | | | | | |
> > > | Qwen3-8B | 35.4 | 24.5 | 14.4 | 13.0 | 6.4 | 6.3 |
> > > | Qwen3-235B | 33.5 | 27.2 | 13.4 | 15.0 | 5.0 | 6.0 |
> > > | R1-Distill-7B | 36.7 | 19.7 | 12.8 | 9.7 | 8.5 | 12.6 |
> > > | *Avg. Thinking* | *34.6* | *21.9* | *14.3* | *11.2* | *7.5* | *10.5* |

---

### Official Review · Reviewer_UJ2U · 2025-10-31

**Soundness:** 4
**Presentation:** 4
**Contribution:** 3
**Rating:** 8
**Confidence:** 5

**Summary:**

This paper evaluates the critical problems of "overthinking" and "underthinking" in long reasoning models. It introduces OptimalThinkingBench, a unified benchmark designed to evaluate the thinking efficiency of these models. The authors conduct a comprehensive evaluation of 33 LLMs on this benchmark, compare the performance of several efficiency optimization methods, and provide a detailed analysis of specific cases. The paper's key finding—that no current model achieves an optimal balance of thinking efficiency—is insightful to this field.

**Strengths:**

1. Comprehensive Workload and Evaluation: The paper demonstrates a substantial amount of work. The benchmark is constructed through automatic synthesis and filtering existing datasets. The experimental section is thorough, covering a broad spectrum of LLMs and testing the effectiveness of multiple optimization methods. The findings and analysis are also detailed and insightful.
2. Sound Metric Design: The proposed metrics avoid subjective semantic judgments of what constitutes "overthinking" and "underthinking". Instead, they rely on objective, quantifiable, and reproducible measures: thinking token count and final answer accuracy. This approach provides a robust and practical standard for evaluation.
3. Scalable and Diverse Benchmark Construction: The benchmark's construction relies on automated synthesis and filtering, ensures it is extensible for future models and helps prevent dataset contamination. Furthermore, the benchmark's broad domain coverage facilitates a more holistic evaluation of model performance across diverse tasks.

**Weaknesses:**

1. Lack of formal definitions: The paper relies heavily on an intuitive understanding of "overthinking" and "underthinking" rather than providing precise, formal definitions. This ambiguity can hinder a clear grasp of the exact problem being solved and makes it difficult to establish firm boundaries for what qualifies as each behavior.

2. Potential for dataset bias in UnderthinkingBench: The construction of the UnderthinkingBench is filtered based on the performance gap between two specific models (Qwen3-1.7B and Qwen3-235B-A22B) . This reliance on a specific pair of models may introduce significant dataset bias, potentially making the benchmark overly tailored to the strengths and weaknesses of these models. Furthermore, the "performance-only" evaluation approach may amplify this bias, the developer can not capture the true nature of the cognitive failure and is less conducive to developing genuine solutions.

3. Limited analysis of root causes: While the paper provides robust metrics for what is happening (evaluation), it offers limited analysis into why models overthink or underthink. The evaluation neither reveal the root causes of these inefficiencies, nor provide clear, actionable directions for model optimization. Given that "overthinking" and "underthinking" are widely discussed problems, a paper introducing a benchmark would be stronger if it also provided deeper diagnostic insights, moving beyond scoring to explain the underlying mechanisms.

**Questions:**

1. There appears to be a discrepancy in the paper. Line 209 states the filtering threshold $\lambda=0.1$, while line 744 states $\lambda=0.3$. Could the authors clarify which value was actually used for the dataset construction?
2. What's the reasoning for selecting $t_{max} = 1000$ as the maximum thinking length for the $AUC_{OAA}$ calculation? The experimental results (e.g., in Table 1) show that several models generate far more than 1000 tokens on average for simple queries.

---

> ### Author Response · Authors · 2025-11-20
>
> We thank the reviewer for their positive assessment and for recognizing our work’s comprehensiveness, sound metric design, and scalable construction. We address the reviewer’s comments and questions below.
>
> We are happy to address your questions and suggestions below:
>
> > Lack of formal definitions: The paper relies heavily on an intuitive understanding of "overthinking" and "underthinking" rather than providing precise, formal definitions...
>
> While we agree that formal definitions would be good to have in future work, we believe our objective quantifiable metrics provide accurate signals of a model’s optimal thinking capability and where it can be improved.
> We operationally define overthinking as generating excessive thinking tokens without accuracy improvement compared to non-thinking models (Section 2.1), and underthinking as achieving lower accuracy than thinking models despite model scale (Section 2.2). This approach avoids subjective semantic judgments while providing reproducible evaluation, as you noted in the strengths. We believe formal definitions would risk being either too restrictive or too abstract to be practically useful across diverse model architectures and domains.
>
> ---
>
> > Potential for dataset bias in UnderthinkingBench: The construction of the UnderthinkingBench is filtered based on the performance gap between two specific models (Qwen3-1.7B and Qwen3-235B-A22B) . This reliance on a specific pair of models may introduce significant dataset bias, potentially making the benchmark overly tailored to the strengths and weaknesses of these models.
>
> We believe the reliance on a model pair is a strength of our recipe, as it allows the benchmark to evolve by switching reference models as the field progresses. We do not believe this biases the evaluation towards specific models; since
> a.) the performance gap between thinking and non-thinking models persists across all model families,
> b.) Neither Qwen3-1.7B nor Qwen3-235B-A22B performs the best (and are often beaten by smaller models)
> We selected Qwen primarily because it is the only open-source family offering comparable reasoning and non-reasoning variants across multiple sizes.
>
> ---
>
> > Furthermore, the "performance-only" evaluation approach may amplify this bias, the developer can not capture the true nature of the cognitive failure and is less conducive to developing genuine solutions.
>
> As noted in your assessment of our strengths, we prioritized metrics that "rely on objective, quantifiable, and reproducible measures" over subjective judgments. However, to capture the nature of cognitive failures beyond just scores, we performed a rigorous qualitative analysis in Section 3.4 and Appendix D.4, detailing specific failure modes such as reasoning loops, switching from a correct answer to an incorrect one due to overthinking, and reliance on heuristics resulting in underthinking.
>
> ---
>
> > Limited analysis of root causes: While the paper provides robust metrics for what is happening (evaluation), it offers limited analysis into why models overthink or underthink. The evaluation neither reveal the root causes of these inefficiencies, nor provide clear, actionable directions for model optimization...
>
> Our analysis does reveal when and where models overthink more on STEM domains regardless of actual complexity (Figure 4a), numerical tokens trigger excessive thinking even for simple facts (Section 3.4), and adding irrelevant MCQ options causes linear increases in overthinking (Figure 10). In addition, we qualitatively analyze reasoning chains and find that excessive thinking often leads to conflicting reasoning and models often rely on heuristcs leading to underthinking.
> While our primary contribution is providing a scalable benchmark generation recipe and establishing objective evaluation standards, we agree that automated mechanistic analysis represents an exciting direction for future work.
>
> ---
>
> > Question: Line 209 states the filtering threshold as 0.1, while line 744 states 0.3. Could the authors clarify which value was actually used for the dataset construction?
>
> Thanks for pointing this out! The correct threshold is \lambda=0.1 (Line 209). We have corrected Line 744 in the revised paper
>
> ---
>
> > Question: What's the reasoning for selecting as the maximum thinking length for the calculation? The experimental results (e.g., in Table 1) show that several models generate far more than 1000 tokens on average for simple queries.
>
> The specific choice of 1000 was based on a reasonable a priori expectation, rather than choosing after inference the models, as adjusting the threshold to fit observed model behaviors would be unfair and biased. We believe that this parameter is a strength of the recipe, allowing adjustment as model thinking evolves to be longer or shorter in the future.
>
> Thank you again for your constructive feedback and positive assessment. We look forward to addressing any other questions you may have!

---

> > ### Comment · Reviewer_UJ2U · 2025-11-24
> >
> > Thank you for your detailed replies. I have a few more points I'd like to discuss.
> >
> > 1. I agree that Overthinking/Underthinking is difficult to formally define, and I also acknowledge the author's concerns about formal definition. However, as a benchmark, a clear task definition is essential (even if it may be biased) and should be correlated with metrics. A clear definition helps readers understand the problem the benchmark aims to solve. Readers can intuitively understand the model's performance on the problem under study from the metrics. My personal view is that evaluations of the thinking process may not be entirely detached from semantics, although this paper makes a very good attempt based on objective metrics.
> >
> > 2. Some studies have pointed out that the Qwen series models may have data leakage[1], which raises my concerns about the construction of the Underthinking Benchmark. In an extreme case, if the Qwen3-1.7B model is simply repeating the correct answer, then this Underthinking Benchmark is actually just constructing a list of tasks that Qwen3-235B-A22B cannot complete. This is why I believe that using only this pair of models for selection may be biased. Consider another approach: Could hybrid thinking models like Qwen3-235B-A22B or DeepSeek-V3.1-Terminus be used for benchmark building? We can easily find many problems where "introducing thinking leads to errors."
> >
> > [1] Wu M, Zhang Z, Dong Q, et al. Reasoning or memorization? unreliable results of reinforcement learning due to data contamination[J]. arXiv preprint arXiv:2507.10532, 2025.

---

> > > ### Author Response · Authors · 2025-12-02
> > >
> > > > I agree that Overthinking/Underthinking is difficult to formally define, and I also acknowledge the author's concerns about formal definition. However, as a benchmark, a clear task definition is essential (even if it may be biased) and should be correlated with metrics. A clear definition helps readers understand the problem the benchmark aims to solve. Readers can intuitively understand the model's performance on the problem under study from the metrics. My personal view is that evaluations of the thinking process may not be entirely detached from semantics, although this paper makes a very good attempt based on objective metrics.
> > >
> > > We thank the reviewer for appreciating our objective metrics and for concurring with our concerns about fully formal definitions. We agree that clear definitions, even if somewhat biased, are helpful for readers, and have therefore added the following to the revised paper (Sections 2.1 and 2.2): 1.) Overthinking: When a model generates more reasoning tokens than necessary to solve a problem that could be solved correctly by the same model with fewer or no reasoning tokens.
> > > 2.) Underthinking: When a model prematurely abandons or insufficiently engages in reasoning steps, resulting in errors that could have been avoided with more deliberate thinking.
> > >
> > > ---
> > >
> > > > Some studies have pointed out that the Qwen series models may have data leakage[1], which raises my concerns about the construction of the Underthinking Benchmark. In an extreme case, if the Qwen3-1.7B model is simply repeating the correct answer, then this Underthinking Benchmark is actually just constructing a list of tasks that Qwen3-235B-A22B cannot complete. This is why I believe that using only this pair of models for selection may be biased. Consider another approach: Could hybrid thinking models like Qwen3-235B-A22B or DeepSeek-V3.1-Terminus be used for benchmark building? We can easily find many problems where "introducing thinking leads to errors."
> > >
> > > We thank the reviewer for clarifying their point. While it is indeed possible that Qwen models are contaminated, the UT-Reasoning split in UnderthinkingBench is algorithmically generated (every question is new), making contamination unlikely. Further, to test whether our model pair introduces bias, we checked whether alternative pairs would select similar tasks. Specifically, we evaluated: a) GPT-4o vs Qwen3-1.7B, b) GPT-4o vs R1-Distill-7B, c) Qwen3-235B-nonthink vs R1-Distill-7B, and d) Qwen3-235B-nonthink vs Qwen3-235B-think (the hybrid model you suggested). Out of our original 11 selected tasks, pairs a), b), c), and d) would independently select 9, 8, 8, and 6 tasks respectively. This high overlap demonstrates that our task selection is not systematically biased towards the original Qwen pair.
> > >
> > > ---
> > >
> > > Once again, we thank the reviewer for their positive assessment of our work, and for engaging in the discussion! We believe the clarifications have further strengthened our work!

---

### Official Review · Reviewer_Vrui · 2025-10-31

**Soundness:** 3
**Presentation:** 3
**Contribution:** 3
**Rating:** 6
**Confidence:** 4

**Summary:**

The paper proposes a benchmark unifying the overthinking and underthinking for LLMs and corresponding metrics to evaluate the LLM performance on accuracy and efficiency.

**Strengths:**

1. The benchmark jointly evaluates LLM performance on both simple and challenging tasks. This assessment reveals whether LLMs can adaptively allocate reasoning effort based on task difficulty, thus achieving an optimal balance between accuracy and efficiency.
2. The paper proposes the overthinking-adjusted accuracy, which presents the token-efficiency of reasoning LLM more precisely
3. The paper conducts experiments on a wide range of LLMs, which provide a comprehensive and solid evaluation of the current model performance.

**Weaknesses:**

1. The underthinking dataset is constructed based on performance differences between small reasoning models and large non-reasoning models. However, the evaluation method and the reason for the selected performance threshold are not clearly explained. Additionally, the construction process does not account for the absolute performance of each model. If the reasoning model also performs poorly on a given task, there could be factors beyond reasoning capability, such as context misunderstanding or knowledge gaps, causing the failures. Performance on such tasks may not precisely reflect reasoning capabilities.
2. The information provided by precision, recall, and F1 scores is limited. Overthinking is a complex behavior that can arise from different underlying causes. The current metrics only assess response accuracy with and without token length constraints, but cannot reveal how responses evolve over time. For instance, on simple questions where LLMs can derive answers in a few tokens, it remains unclear how the model behaves after generating the correct answer. Understanding this evolution is crucial for improving LLMs' optimal thinking capabilities.

**Questions:**

1. What is the advantage of using this unified benchmark compared to combining two existing benchmarks that separately target overthinking and underthinking?

---

> ### Author Response · Authors · 2025-11-20
>
> We thank the reviewer for their thoughtful review and appreciation of our unified benchmark, the proposed overthinking-adjusted accuracy metric, and our comprehensive evaluation.
>
> We are happy to address your questions and suggestions below:
>
> > The underthinking dataset is constructed based on performance differences between small reasoning models and large non-reasoning models. However, the evaluation method and the reason for the selected performance threshold are not clearly explained
>
> For bench creation, we use the standard evaluation method for individual tasks as used in the final evaluations, which have been detailed in Section 2.2 (Lines 221-228). Further, for each task, we evaluate 50 questions on 8 random seeds and take the average. We have clarified this in the revised paper.
>
> ---
>
> > reason for the selected performance threshold
>
> We currently set the threshold to 0.1 to obtain a representative set of tasks satisfying the principle that "for certain questions, no matter how large a non-thinking model is, its performance on complex reasoning tasks will be lower than that of a much smaller yet thinking model," while also accounting for statistical uncertainty in estimating model performance. We have clarified this in the revised paper. Beyond the benchmark itself, one of our contributions is a general recipe for creating an underthinking benchmark where the threshold allows for flexibly adjusting the difficulty of the benchmark as needed, as models evolve, and hence should be seen as a strength of our recipe.
>
> ---
>
> > Additionally, the construction process does not account for the absolute performance of each model. If the reasoning model also performs poorly on a given task, there could be factors beyond reasoning capability, such as context misunderstanding or knowledge gaps, causing the failures. Performance on such tasks may not precisely reflect reasoning capabilities.
>
> This is a great point, and it is the precise reason why our `UT-Reasoning' split is derived from Reasoning Gym (Table 5). These tasks are algorithmic (e.g., games, arithmetic), where the method for solving the problem is self-contained or well-known, unlike factual QA tasks. Therefore, poor performance on this benchmark is primarily driven by a lack of sufficient reasoning (underthinking) rather than context misunderstanding or knowledge gaps.
>
> ---
>
> > The current metrics only assess response accuracy with and without token length constraints, but cannot reveal how responses evolve over time. For instance, on simple questions where LLMs can derive answers in a few tokens, it remains unclear how the model behaves after generating the correct answer.
>
> Our goal with OptimalThinkingBench is to provide an objective and efficient method to jointly evaluate overthinking and underthinking, unlike previous works that have often studied them qualitatively and in isolation. However, to address the evolution of responses, we do qualitative analysis in Section 3.4 and Appendix D.4. Specifically, Listings 1, 2, and 3 demonstrate overthinking where the "model initially comes up with the correct answer but then overthinks either because of conflicting information or incorrect reasoning" (Lines 1216-1217). Conversely, Listings 4 and 5 show underthinking where models rely on heuristics without self-verification. W believe developing automated metrics for this evolution is an exciting direction for future work.
>
> ---
>
> > Question: What is the advantage of using this unified benchmark compared to combining two existing benchmarks that separately target overthinking and underthinking?
>
> The primary advantage is standardization: existing studies often use disparate evaluation setups and metrics (Section 4), "making fair comparison across approaches difficult and hindering systematic progress" (Lines 468-469). Furthermore, a unified benchmark is critical because optimizing for one metric often degrades the other. As shown in our experiments (Section 3.3), existing efficiency methods reduce overthinking but harm performance on complex tasks; our unified $F_1^{\text{otb}}$ metric ensures that models are penalized unless they can adaptively allocate compute based on task difficulty, i.e, they think "optimally".
>
>
> ---
>
> Thank you again for your thoughtful review. We hope our responses have addressed your concerns. If you think there are still some concerns, we look forward to addressing them as well! Alternatively, if you feel all your concerns are addressed, we would appreciate it if you could reflect that in your evaluation!

---

### Meta-Review · Area_Chair_ftnq · 2026-01-05

**Summary:**

**Summary**:
The paper addresses the critical trade-off between efficiency and performance in modern "thinking" LLMs (e.g., o1, DeepSeek-R1). It introduces "OptimalThinkingBench," a unified benchmark composed of two sub-benchmarks: "OverthinkingBench" (simple queries where efficiency is key) and "UnderthinkingBench" (complex tasks requiring deep reasoning). The authors propose a unified $F_1$ score derived from Overthinking-Adjusted Accuracy ($AUC_{OAA}$) and standard accuracy. By evaluating 33 models, the paper demonstrates a "seesaw effect" where current models fail to optimally balance this trade-off, tending to either overthink on trivial tasks (wasting compute) or underthink on complex ones (failing to reason).

**Rebuttal Conclusion**:
The authors provided an exceptionally strong rebuttal that effectively addressed the reviewers' concerns, particularly regarding dataset construction validity. While Reviewer gas8 (Score: 4) remained negative, the AC has chosen to discount this opinion due to factual inaccuracies in their initial review (e.g., claiming missing baselines that were actually present). Conversely, Reviewer UJ2U (Score: 8) identified a critical potential flaw regarding data leakage in the "Underthinking" subset but was fully satisfied by the authors' new cross-model validation experiments. The consensus is that the paper presents a timely, rigorous, and necessary benchmark for the community as inference-time scaling becomes a central research topic.

**Reviewer Concerns:**

**Concerns Addressed by the Rebuttal**:
•**Data Leakage and Bias in Benchmark Construction**: Reviewer UJ2U raised a critical concern that the "UnderthinkingBench," constructed using Qwen model pairs, might suffer from data leakage or model-specific bias. The authors resolved this by performing cross-model validation (using GPT-4o, R1-Distill, etc.) to re-select tasks. The results showed high overlap in task selection, proving the benchmark captures general reasoning difficulties rather than model artifacts.
•**Missing Baselines and Methods**: Reviewer gas8 initially rejected the paper claiming a lack of commercial model evaluation and adaptive routing methods. The authors resolved this by pointing to specific Tables (1, 2, and 3) where these exact models (GPT-4o, o3) and methods (Dynamic Routing) were already evaluated.
•**Metric Limitations ("Black Box")**: Reviewer Vrui noted that the metrics do not capture the temporal evolution of thought (e.g., self-correction). The authors addressed this by adding a rigorous qualitative analysis and a new quantitative analysis (Appendix D.5) categorizing underthinking behaviors (e.g., lack of verification), which complements the outcome-based metrics.
•**Threshold Definitions**: Reviewer Vrui questioned the logic behind the filtering thresholds. The authors clarified that the thresholds are statistically grounded and the task nature (puzzles/algorithms) minimizes knowledge-based variance.

**Outstanding Concerns**:
•**Process Interpretability**: Reviewer Vrui noted that while the metrics are robust for ranking, they remain outcome-based and do not fully reveal why a model overthinks or underthinks at a granular cognitive level. The AC agrees this is a limitation but views it as an avenue for future interpretability work rather than a flaw in a benchmark paper.
•**Extent of Underthinking Analysis**: Reviewer gas8 requested further analysis on the "extent" of underthinking. While the AC believes the authors' added experiments in Appendix D.5 are sufficient, the reviewer remained lukewarm.

**Reviewer Scores:**

I think the estimated final sentiment is split:4,6,8.
While UJ2U and Vrui support acceptance, gas8 remains an outlier. The AC discounts this review due to factual inaccuracies (e.g., overlooking existing baselines), though the score increase (2 to 4) reflects recognition of the rebuttal's effectiveness.

---

### Decision · Program_Chairs · 2026-01-26

Accept (Poster)